# Machine learning assisted single-molecule sensing towards standard-free quantification of per- and polyfluoroalkyl carboxylic acids

Jiaqi Zuo [1,2,5], Hong-Shuang Li [1,5], Wen Tang[1,5], Xian Zhao[1], Meng-Yuan Cheng[1], Zekai Yang[1], Siyu Tian[1], Pufeng Li[1], Xueying Xie[1], Dan Luo[1] & Kaipei Qiu [1,3,4] ✉

Per- and polyfluoroalkyl carboxylic acids (PFCAs) are of global concern for their ubiquitous presence in the environment. However, precise quantification of PFCAs remains challenging due to the shortage of standards. Herein, with the aid of machine learning, a probe-directed nanopore based single-molecule electrochemical sensor is developed towards standard-free digital quantification of PFCAs. To correctly predict the signal without standards, a strict linear relationship ($R^2 > 0.9998$) is established between current blockades and molecular volumes of PFCAs up to C14. Leveraging high-resolution multi-feature classification, identification accuracy reaches 100% for a broad range of PFCAs including isomers. Reliable, multiplexed quantification of PFCAs is verified in various environmental matrices, with a state-of-the-art detection limit of 0.1 nM for trifluoroacetic acid (an ultrashort-chain PFCA). The double-barriers of probe-pore interaction suggest that capture rates can be independently tuned, without comprising identification. As a proof-of-concept, a universal probe-determined calibration curve is realized experimentally for short- and medium-chain PFCAs, which is theoretically extendable to all PFCAs for standard-free quantification via nanopore engineering.

Per- and polyfluoroalkyl substances (PFAS) contamination has become a global concern due to their widespread environmental occurrence and elevated human exposure[1]. It is a complex chemical class, covering almost all the compounds with at least a perfluorinated methyl (-CF$_3$) or methylene (-CF$_2$-) functional group, according to the revised definition by the Organisation for Economic Cooperation and Development (OECD)[2]. Till now, over 14,000 chemicals have been registered in the U.S. EPA PFAS structure list[3]. Growing evidence indicates that the abundant tiny structural variation in PFAS can affect the respective partitioning[4], transferring[5], elimination[6], transformation[7], bioaccumulation[8] behaviors, and may pose distinct health risk[9]. Hence,

it is imperative to determine the concentration of multiple structurally similar PFAS species in various environmental matrices simultaneously.

The current analytical methods for PFAS, however, are hard to achieve precise quantification, high resolution, and wide coverage concurrently. While great advances have been made over the past decades to enhance the resolution and detection limit of high-performance liquid chromatography-tandem mass spectrometry (HPLC-MS/MS)[10,11], as well as gas chromatography-mass spectrometry (GC-MS)[12,13], such quantitative analysis relied on the use of reference standard, of which the number of commercially available samples[14]

[1]Key Laboratory of Environmental Risk Assessment and Control on Chemical Process, Ministry of Ecology and Environment, School of Resources and Environmental Engineering, East China University of Science and Technology, Shanghai, PR China. [2]State Key Laboratory of Estuarine and Coastal Research, East China Normal University, Shanghai, PR China. [3]State Key Laboratory of Coal Liquification, Gasification and Utilization with High Efficiency and Low Carbon Technology, Shanghai, PR China. [4]Shanghai Institute of Pollution Control and Ecological Security, Shanghai, PR China. [5]These authors contributed equally: Jiaqi Zuo, Hong-Shuang Li, Wen Tang. ✉e-mail: kaipeiqiu@gmail.com

were just over 120, less than 1% of the total PFAS. In contrast, progress in high-resolution mass spectrometry (HRMS)[15,16] opens the possibility to comprehensive, nontargeted screening of unknown PFAS, especially when coupled with ion mobility spectrometry (IMS)[17,18]. Nevertheless, the structure identified by HRMS is tentative[19], and the concentration can, at best, be semi-quantified. So far, quantification of PFAS without standards remains challenging.

Herein, a nanopore based single-molecule electrochemical sensor is proposed as an emerging technology to bridge the gap between the urgent demand for quantitative PFAS monitoring and the severe shortage of authentic standards. By measuring the change of ion flow through nanopore, single-molecule sensor (SMS) can identify and quantify specific analytes with the characteristics and frequency of resulting current blockade[20]. To avoid the use of standards in identification, peptide probes were designed to direct the movement of PFAS in nanopore so that a linear correlation was formed between the magnitude of ionic current and the volume of PFAS simulated by molecular dynamics (MD), and the current response of other unknown PFAS could be accurately predicted. Previous studies have demonstrated positive correlations between the magnitude of current blockade and the volume[21] or mass[22-26] of peptides[21-24] and proteins[25,26] using a variety of protein pores, e.g., α-hemolysin[22], aerolysin[21], FraC[23], ClyA[25], CytK[24] or YaxAB[26], but a strict linear relationship ($R^2 > 0.9998$) had not been realized yet, as far as the authors were aware, in which the predicted blockade values were almost identical to experimental measurements. To further enhance the resolution of PFAS identification, a frequency-modulated multi-dimensional feature extraction-based machine learning algorithm was developed, reaching an overall accuracy of 99.9% for a total of 13 PFCAs. Further optimization of feature combination, reducing it from 43 to 21 dimensions, required only 13% of test set for 99% accuracy. As a result, even in an interference of 100 times concentration, nanopore SMS was able to maintain 78% identification reliability, over an order of magnitude better than ensemble analysis. Simultaneous quantification of multiple PFAS was facile based on the calculation of component-specific interval time[27], as the overall capture rate observed was the sum of partial rates of individual analytes (analogous to ideal mixture), and common interference in the environmental matrices had a negligible effect on single-molecule quantification of PFCAs. Most importantly, it was demonstrated that the capture rates of short- and medium-chain PFCAs were determined by the tethered polycationic peptide probes, and could be independently tuned without compromising the linear blockade-volume relationship thanks to the double-barrier of probe-nanopore interaction, offering an opportunity to pursue standard-free quantification in the future. As proof of concept, a wide, interference-free linear-response range of 0.1 nM to 100 μM was achieved for trifluoroacetic acid, corresponding to the detection limit of 11.4 ng L$^{-1}$, which was comparable to the state-of-the-art performance of UPLC-MS/MS[28] or GC[29].

## Results
### Establishing a linear volume-blockade relationship for PFCAs
Building a structure-activity relationship was the prerequisite towards the development of standard-free quantification methods, and thus a linear correlation was established in the first place between the volume of PFCA molecules and the magnitude of current blockade measured by nanopore SMS. Despite many factors were proposed to influence the translocation induced current response, it was possible to compile them into two dominant factors, i.e., steric exclusion, counterion enhancement, or a combination of those two[30]. As for steric exclusion, the effective volume of nanopore for signal transduction could change dynamically, indicating that the residence position of analyte in nanopore was crucial when sensing small molecules[31]. Therefore, polycationic peptide probes were employed in this study to control the location of tethered PFCAs in nanopore, while concentrated

electrolytes were adopted to reduce the contribution of surface charge, so that the magnitude of current blockade was mainly determined by steric exclusion. More specifically, eight linear perfluoroalkyl carboxylic acids (C2 to C9) were chosen as typical PFCAs to form the structure-current relation, Fig. 1a. They were connected to the N-terminal of an oligo-arginine leader (PFCA-R$_6$) and measured by the wild-type aerolysin (WT AeL) in 4 M KCl solution, Fig. 1b. It was found previously that WT AeL formed positive electrostatic barriers on both of its trans exit and cis entry under negative applied voltages[31], and so the polycationic -R$_6$ probe might be able to drive the non-ionic PFCA targets to the identical position within the AeL nanopore. The typical current traces of C2-C9 PFCA-R$_6$ at −50 mV were shown in Fig. 1c, as well as the R$_6$ probe (C0). It was clear that both the magnitude of current blockade and dwell time increased with the length of PFCA. Histograms of the current blockade for C0 and C2 to C9 were also given in Supplementary Fig. 1. It was worth noting that, for PFAS that didn't contain the carboxyl terminal group, it was facile to design other types of probes with different recognition domains.

The resulting magnitude of current blockade for C2-C9 PFCA-R$_6$ complexes exhibited a strong linear correlation ($R^2 > 0.9998$) with their molecular volume (shown as shallow squares in Fig. 1d). The measured differences between the blockade of C2- and C9-R$_6$ was 11.8%, corresponding to an increase of 1.68% per -CF$_2$- (ca. 73.5 Å$^3$) or a slope of 0.023% Å$^{-3}$ for this straight line. The effective transduction volume of 4.82 nm$^3$, estimated at the blockade of 100%, was identical to the inner pore volume comprised between the A224 and S236 residues of WT AeL (4.82 nm$^3$), supporting the use of R$_6$ probe to direct the movement of PFCA targets. A total of 61 individual measurements (at least three for each sample) were conducted to reduce experimental errors, using perfluorohexanoic acid as an internal standard for calibration[32]. The average of error between different measurements was as low as 0.022%, one order of magnitude smaller than the overall standard deviation (0.198%) for the histogram of current blockade, confirming the reliability of our approach. The hydrodynamic volume of PFCA-R$_6$ was calculated via all-atom molecular dynamics simulations using GROMACS. More details of the simulation process were summarized in Methods, and the obtained raw data of volumes for C0 and C2 to C9 were given in Supplementary Figs. 2 and 3.

### Predicting the current blockade of H- and Cl-substituted PFCAs
Following the establishment of a linear correlation between the volume of perfluoroalkyl carboxylic acids and their magnitude of current blockade, the next step was to examine its prediction accuracy for other polyfluoroalkyl carboxylic acids. Among all the 622 carboxylic PFAS structures that were identified previously (CAS numbers available),[5] more than 70% fell into the volume range of C2 to C9, while 49% contained at least a C-H functional group, and 5.7% had one or more C-Cl groups. Therefore, five typical H- or Cl-substituted analytes, either terminal or internal, were examined in this study, including 3H-tetrafluoropropionic acid (3H), 5H-octafluoropentanoic acid (5H), 7H-dodecafluoroheptanoic acid (7H), 3Cl-tetrafluoropropionic acid (3Cl), and 3:3 fluorotelomer carboxylic acid (FTA), which were increasingly discovered in the wastewater from fluorochemical industry as well as the surrounding surface water[33-35]. Based on the MD simulated molecular volume, the predicted current blockades of H- or Cl-PFCAs were in perfect accordance with experimental measurements (Fig. 1d and insert), with negligible deviation (0.022%) close to the observational errors, which clearly demonstrated the capability of using nanopore SMS for standard-free prediction of PFAS. The current trace, histogram of blockade, and the simulated molecular volume of H- or Cl-PFCAs were provided in Supplementary Figs. 4 and 5. In addition to the above, five more other types of PFCAs were examined (Supplementary Fig. 6), including one with a methyl side-chain (3,3,3-trifluoro-2-methylpropanoic acid), another one with a methyl side-chain and an unsaturated bond (3-(trifluoromethyl)crotonic acid), one with a hydroxyl side-

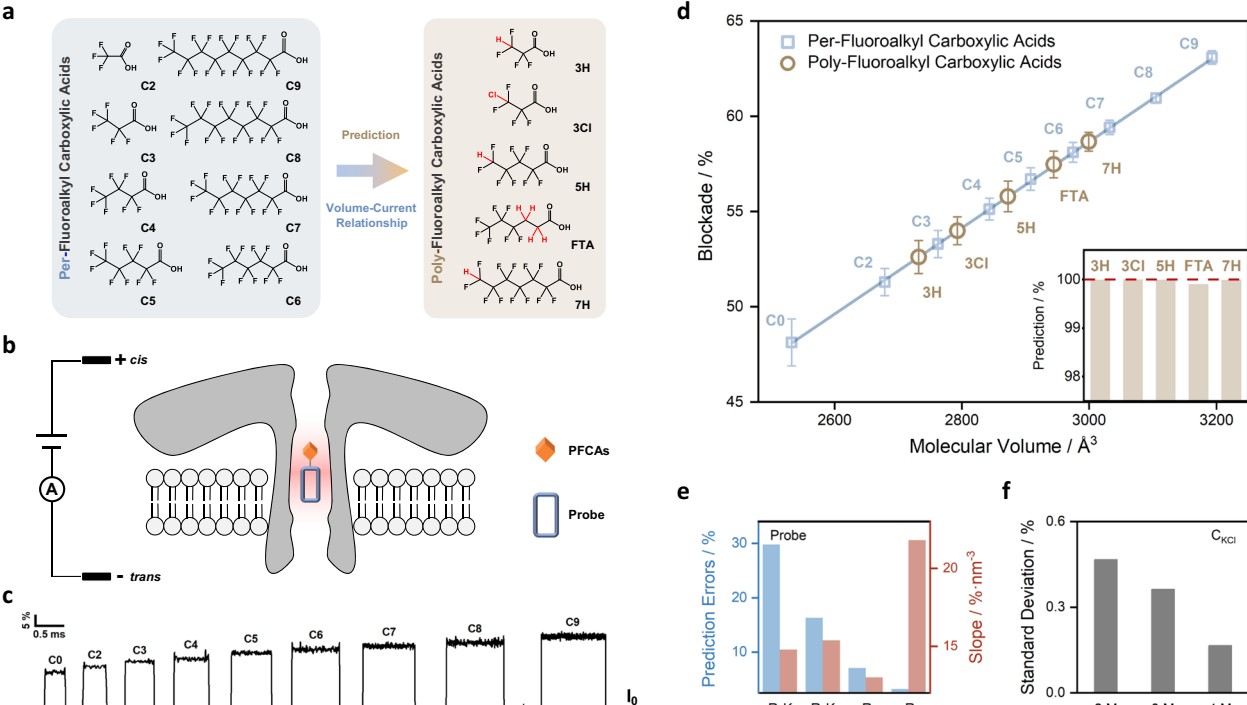

**Fig. 1 | Establishment of a linear volume-current relationship for standard-free prediction of unknown signal in single-molecule sensing of PFCAs. a** The eight per-fluoroalkyl carboxylic acids used in this study to form the linear volume-current correlation (blue box on the left), and the five H- or Cl-substituted poly-fluoroalkyl carboxylic acids adopted to validate its correctness (brown box on the right). **b** The PFCA molecules were tethered to four kinds of polycationic probes, -R$_6$, -R$_7$, R$_5$K- and R$_6$K-, and were measured by WT AeL nanopore at −50 mV in 4 M KCl. **c** Typical current traces measured in the single-molecule sensing of C2- to C9-R$_6$, as well as the R$_6$ probe (C0). **d** The established linear relationship between the hydrodynamic volume of C0/C2-C9 PFCA-R$_6$ and the magnitude of their current blockade (blue squares and straight line) (linear fit, $R^2 > 0.9998$). The volumes of PFCAs were

calculated via MD simulation using GROMACS (Supplementary Figs. 2 and 3). The experimentally measured current blockade of 3H, 5H, 7H, 3Cl, and FTA was shown as brown circles, which was almost identical with the prediction from volume-current correlation (insert in **d**). The error bars on blue squares and brown circles were the sum of three times standard deviations (NOT one) obtained from the histograms of their current blockades, which were recorded from a total of 61 individual measurements (at least three for each sample). **e** Prediction errors for the current blockade of C6 using R$_6$K-, R$_5$K-, -R$_7$, or -R$_6$ probe (blue bars), and the slope of their volume-current response (red bars). **f** The average standard deviation for the histograms of current blockade of C5-C7 PFCA-R$_6$ measured in 2, 3, and 4 M KCl solutions.

chain ((R)-3,3,3-trifluoro-2-hydroxypropanoic acid), and two with aromatic groups (2,3,4,5-tetrafluorobenzoic acid and 2,6-bis(trifluoromethyl)benzoic acid), confirming the wide applicability of the volume-current relation established in this work.

### Factors affecting standard-free prediction and identification

To further explore the origin of the observed linear correlation, different peptide structures (R$_6$K-, R$_5$K-, -R$_7$ and -R$_6$) were compared to analyse the role of probe length and orientation of connection. Perfluoropentanoic (C5), perfluorohexanoic (C6) and perfluoroheptanoic acid (C7) were linked to the lysine side chain of R$_6$K- or R$_5$K- probes, and to the N-terminals of -R$_7$ or -R$_6$. Errors of the actual blockade from prediction was much smaller for C6-R$_6$ or -R$_7$ than R$_6$K- or R$_5$K-C6 (Fig. 1e), probably due to the narrow lumen of WT AeL (diameter between 1 and 1.4 nm)[36]. Meanwhile, the slope of the linear fit for -R$_6$ probe, i.e., signal sensitivity to the change of analyte volume, was 70% higher than -R$_7$ (Fig. 1e), suggesting the enlarged transduction volume of the latter. Although showing little improvement in linearity ($R^2 = 0.9995$-0.9999 for the linear fit of C5-, C6-, and C7-R$_6$ in 2–4 M KCl (Supplementary Fig. 7), the elevated salt concentration reduced the standard deviation of blockade from 0.47% for 2 M to 0.165% for 4 M (Fig. 1f), which could triple the resolution of identification. One possible reason was the prolonged dwell time of PFCAs in WT AeL, caused by the higher cis-to-trans driving force in 4 M KCl (or the lower trans-to-cis electroosmotic force)[20]. Nevertheless, it was noted that the sum of the three sigma of poly- and the adjacent per-fluoroalkyl carboxylic acids (highlighted as the error bar in Fig. 1d) was still greater than the

difference between their blockade, indicating that the use of current blockade alone was unlikely to fully resolve the total 13 PFCAs.

### Frequency-modulated multi-dimensional feature extraction

An inherent advantage of SMS over ensemble methods was the ability to record multi-dimensional features of the signal generated by an individual molecule[37], rather than the properties of functional groups only. The use of five or more signal features was demonstrated recently as an effective approach to distinguish structurally similar compounds, e.g., achieving 92.4%-99.9% accuracy for the determination of saccharides[38], riboses[39], alditols[40] or benzenediols[41]. Herein, frequency modulation (using five low-pass filters of 2000, 800, 500, 200 or 100 Hz, as well as the wavelet transform) was applied to extend the eight common features of single-molecule signals, i.e., the magnitude ($\Delta I/I_O$), duration ($\tau_{on}$), standard deviation ($I_\sigma$), peak-to-peak ($I_{pp}$) of current blockade, and the peak ($H_{peak}$), full width at half maximum ($H_{FWHM}$), skewness ($H_{skew}$), kurtosis ($H_{kurt}$) of the all-points histograms of current blockade, to a total number of 43 ($\tau_{on}$ remained constant at all frequencies), Fig. 2a. All the feature inputs were normalized by an internal standard (C6, C5 or C3) and averaged by at least three parallel measurements to minimize experimental errors. The resulting 43 features of 14 analytes were all fitted with a Gaussian distribution (Supplementary Figs. 8–21). Interestingly, despite the closely related physical meaning of $\Delta I/I_O$ and $H_{peak}$, or $I_\sigma$, $I_{pp}$ and $H_{FWHM}$, the resolving power of these features and their frequency-dependent behaviors differed remarkably (Supplementary Figs. 22–30), implying the possibility to integrate multi-dimensional features for enhanced resolution.

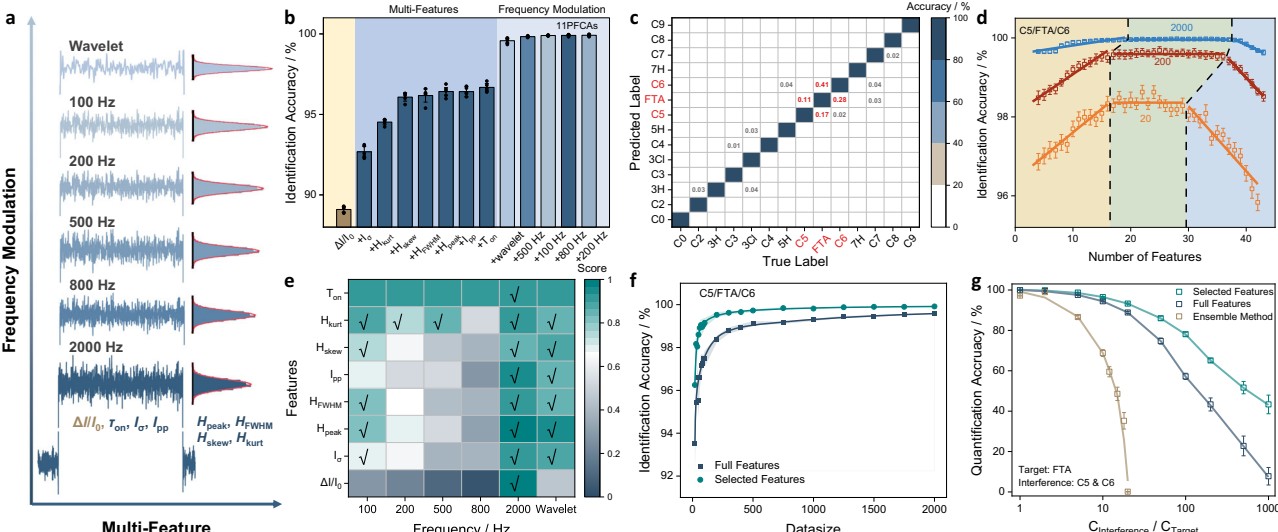

**Fig. 2 | Development of frequency-modulated multi-feature classification for standard-free identification in single-molecule sensing of PFCAs. a** Schematic illustration of frequency-modulated multi-feature extraction approach. The eight features were magnitude ($\Delta I/I_0$), duration ($\tau_{on}$), standard deviation ($I_\sigma$), peak-to-peak ($I_{pp}$) of current blockade, and peak ($H_{peak}$), full width at half maximum ($H_{FWHM}$), skewness ($H_{skew}$), and kurtosis ($H_{kurt}$) of all-points histograms of current blockade. Frequency modulation included wavelet denoising and low-pass filters of 100, 200, 500, 800, and 2000 Hz. **b** Identification accuracy of 11 PFCAs (C2-C7, 3H, 5H, 7H, 3Cl, and FTA) with various feature combinations. Order of feature addition was based on their importance, i.e., to achieve the highest accuracy at any given dimension. The first eight dimensions of feature referred to their values at 2000 Hz, while the subsequent addition was the groups of seven features (except $\tau_{on}$) at different frequency. **c** Confusion matrix for the identification of all 13 PFCAs plus the $R_6$ probe using a bagged decision tree model. For each analyte, 2000 sets of features were used for training and 10-fold cross-validation, and another 400 signals were selected for test. The likelihood of misclassification for the group of FTA/C5/C6 was highlighted in red, and the misidentification that could have been resolved by blockade was labeled in gray. **d** The relationship between the identification accuracy (validation) of FTA/C5/C6 and the number of features adopted, at the training data size of 20, 200 and 2000 for each analyte, respectively (linear fit, $R^2 = 0.8485$-$0.9860$). **e** Importance score of 43 features for the identification of FTA/C5/C6. The ticked ones were used in the shortlisted 21-feature model. **f** The identification accuracy of FTA/C5/C6 against the number of test data using the full 43-feature or shortlisted 21-feature model (exponential fit, $R^2 = 0.9680$-$0.9888$). **g** The identification accuracy of FTA in presence of C5/C6 by ensemble analysis (mimicked by the multi-peak fitting using blockade only), the full 43-feature model, and the shortlisted 21-feature one. The concentration ratio of FTA to the C5/C6 interference ranged from 1:1 to 1:1000. The error bars in (**b**, **d**, **g**) were calculated from at least five parallel tests.

In fact, the combination of eight-dimensional feature (extracted from the raw data at 2000 Hz) increased the identification accuracy from 89.1% (using $\Delta I/I_0$ only) to 96.7% for the 11 short-chain PFCAs (excluding C8 or C9), and the further incorporation of frequency modulation (using the total 43 features) pushed it to 99.9%, Fig. 2b. A total of 2400 sets of data were collected for each analyte (2000 for training/validation and 400 for test). In total, 31 classifiers were evaluated, among which the Bagged trees model showed the highest identification accuracy, Supplementary Table 2. All the accuracies were calculated based on five or more repetitions of random holdout test sets, and a 10-fold cross-validation was applied for training. It was worth noting that the order of feature addition, from two to eight dimensions, for achieving the highest identification accuracy (left part of Fig. 2b), was distinct from the rank of their own one-dimensional accuracy (Supplementary Fig. 31), which suggested the significance of correlation and complementarity among features. The enhancement by feature addition reached a plateau when the number of dimensions exceeded four, meaning that certain features were less relevant or easier to replace in the classification of PFCAs. Once frequency modulation was included, the identification accuracy quickly jumped over 99% (right part of Fig. 2b). The confusion matrix of all 13 PFCAs plus the $R_6$ probe (with an overall accuracy of 99.9%) showed that the biggest errors came from the misclassification of FTA with C5 and C6, Fig. 2c, and strikingly, small but non-negligible false identifications (0.01–0.04%) were constantly observed for the pairs of PFCA that could be resolved completely using blockade only (Supplementary Fig. 23). Such phenomena stressed the necessity to reduce model complexity or the number of features.

## Shortlisting high-priority features for precise SMS of PFCAs
To optimize the combination of model input, the change of maximal validation accuracy against the number of features was examined, under various sizes of training set for C5, C6, and FTA, i.e., 2000, 200, or 20 signals for each PFCA. A similar trend was discovered in all three curves (Fig. 2d): when more features were included, the accuracy increased initially, then reached a plateau, and dropped slightly in the end, indicating that the optimal number of features should be neither too small (e.g., to avoid the deviation in training set) nor too large (e.g., to reduce the model complexity). The height, length, and position of the plateau became lower, shorter, and left-shifted for smaller set, which decreased from 99.97% for 2000 signals with 28 ± 10 features, to 98.40% for 20 signals with 21 ± 5 features. In the meantime, the rank of features was also affected by the size of dataset: when only 20 signals were used, a general order of importance 2000 Hz <wavelet <100 Hz <200 Hz <500 Hz <800 Hz was followed; but in the case of 200 or 2000 signals, other features such as kurtosis at 200 and 500 Hz stood out, while all the filtered blockades (either wavelet or 100–800 Hz) were no longer crucial, Supplementary Tables 3–5. Taking into account of their priority in all three scenarios, a total of 21 features were shortlisted (Fig. 2e), which was able to decrease the amount of test data by 7.6-fold, compared to the full 43-feature model, and to increase the maximal identification accuracy from 99.58% to 99.92% (Fig. 2f).

## Reliable quantification with high-concentration interference
The aforementioned few-shot learning mode of nanopore SMS facilitated the accurate determination of trace targets in the presence of structurally similar interferents with much higher concentrations, which was often challenging in ensemble measurements. For instance,

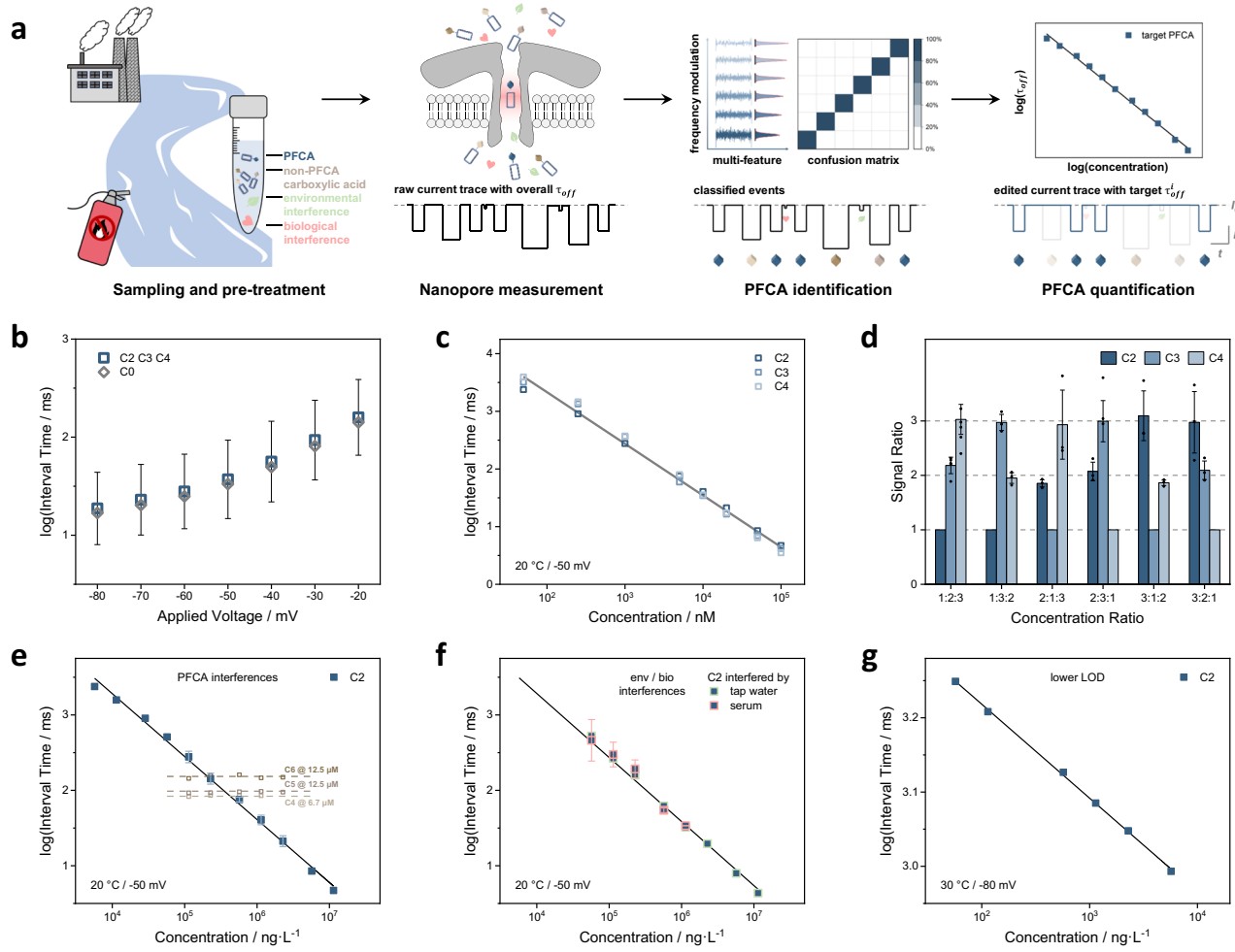

**Fig. 3 | Design of polycationic probes directed capture for standard- and interference- free quantification in single-molecule sensing of PFCAs. a** The unified workflow for probe-directed single-molecule sensing of PFCAs with the assistance of machine learning. **b** Voltage-dependent capture rates for C2, C3 and C4, as well as the $R_6$ probe, under the same concentration. **c** Universal calibration curves for C2, C3 and C4, measured at 20 °C and −50 mV (linear fit, $R^2 = 9911$). The error bars were the average standard deviations obtained from the histograms of interval times of C2, C3, and C4. **d** Comparison of signal ratios for C2/C3/C4 mixtures with molar ratios of 1:2:3, 1:3:2, 2:1:3, 2:3:1, 3:1:2, and 3:2:1, respectively. The number of signals for the fewest PFCAs was normalized to 1. Three repeating

measurements were performed for each mixture. The error bars were standard deviations calculated from at least three parallel measurements. **e** Calibration curves of C2 under the interference of C4, C5 or C6 (linear fit, $R^2 = 0.9963$). The concentration of C2 varied from 50 nM to 100 μM, while those of C4, C5, and C6 were kept at 6.7, 12.5, and 12.5 μM, respectively. **f** Calibration curves of C2 in the presence of serum or tap water (linear fit, $R^2 = 0.9959$). The concentration of C2 varied from 500 nM to 100 μM. The error bars were standard deviations of interval times measured under different interferences. **g** Calibration curve of C2 measured at 30 °C and −80 mV to lower down the limit of detection (linear fit, $R^2 = 0.9993$). The concentration of C2 varied from 0.5 to 50 nM.

with regard to the detection of FTA from more concentrated C5 and C6, the pre-developed 21-feature model was able to maintain over 78% accuracy with interference of 100 times concentration, Fig. 2g, and a substantial 44% was still detectable even under 1000 times. Such performance was at least one order of magnitude better than the multi-peak fit using the 2000 Hz blockade only, which was the best one-dimensional feature in this study to mimic ensemble analysis, showing an accuracy of 69% with 10 times interference and a rapid drop to zero at 20 times. In the meantime, the identification reliability of the shortlisted 21-feature model consistently outperformed the full 43-feature counterpart, in accordance with the above analysis on the influence of feature number.

## Process for standard-free SMS of PFCAs in real-world samples
Combining the probe-directed standard-free volume-blockade prediction and high-resolution multi-feature classification, the standard-free identification of each individual single-molecule signal was readily achievable, and the remaining task was standard- and interference-free

digital quantification of multiple PFCAs by counting their respective number of signals present in a certain period of time. A unified work-flow for single-molecule sensing of PFCAs in environmental matrices included four steps, Fig. 3a: (i) PFCAs were collected, activated, and tethered to polycationic peptide probes (e.g., -R₆), together with other non-PFCA carboxylic acids and various kinds of environmental/biolo-gical interference. To facilitate the separation or promote the con-centration of PFCA-probe conjugates, it was also possible to apply probe-functionalized magnetic beads[42]; (ii) The mixture of (non-)tethered analyte was then transferred to nanopore measurements. Internal standards of known concentrations (e.g., C3-, C5-, or C6-R₆) were added to calibrate the signal feature of identified PFCAs. Since only the capture and/or translocation of probe-connected PFCAs, as well as other non-PFCA carboxylic acids, could cause substantial changes in current, i.e., in terms of blockade magnitude or dwell time, most of the untethered environmental or biological interference would not affect the measurements. (iii) Accurate, simultaneous identification of multiple PFCAs was accomplished by single-molecule

classification based on our in-house library of signal features, of which the construction process did not rely on authentic standards either. As for the probe-connected non-PFCA carboxylic acids (e.g., fatty acids), even when they might possess similar volumes and blockades with PFCAs, their other signal features (e.g., duration or standard deviation of current response) were considerably different in most cases, and thus could be easily distinguished by clustering (Supplementary Fig. 32). Detailed protocols for the single-molecule identification and standard-free preparation of feature library were summarized in Supplementary Methods 1–2; iv) Finally, the number of signals identified for certain PFCAs within a given amount of time was employed to calculate their own interval time, which was inversely proportional to analyte concentration.

### Peptide-directed capture for standard-free quantifications

To perform standard-free quantification, the calibration curves of various PFCAs had to be universal. For a nanopore SMS, the relationship between interval time and analyte concentration was decided by the capture rates of target species. Here, in this particular case, the driving force on probe-PFCAs conjugates was mainly composed of the cis-to-trans electrostatic force on polycationic peptide probe ($R_6$), subtracting the minimized trans-to-cis electroosmotic force. Hence, the captures of PFCAs-$R_6$ could be probe-controlled or analyte-independent if the entry barrier into AeL nanopore was trivial, compared to driving forces. Indeed, it was found that the applied voltage-driven capture patterns of C2-, C3- and C4-$R_6$ (from -20 to -80 mV, for the same analyte concentrations) were almost identical to that of the $R_6$ probe (Fig. 3b and Supplementary Fig. 33), and their corresponding calibration curves (from 50 nM to 100 μM, Fig. 3c and Supplementary Fig. 34) overlapped with each other. To further investigate whether this probe-directed strategy could lead to a universal capture rate for ultrashort- or short-chain PFCAs, mixture of C2-/C3-/C4-$R_6$ with six different combinations of molar ratios was measured (three repetitions for each combination). By normalizing the number of signals to 1 for the analytes of the lowest concentration, the resulting numbers of signals for the other two were very close to 2 and 3, consistent with their concentration ratios (Fig. 3d), which implied that one calibration curve might fit all different kinds of PFCAs.

### Multiplex quantification of PFCAs in environmental matrices

Since nanopore single-molecule sensing of PFCAs was commonly conducted in ideal dilute solution, it was straightforward to implement multiplexed digital quantification by adding up their individual capture rates. To prove it, calibration curves of 10 PFCAs were measured as an example with internal standards of fixed concentrations (Supplementary Figs. 35–39): in this regard, the calculated interval times of internal standards remained constant throughout the process, and were used to re-scale the interval times of target PFCAs, reducing experimental error; more crucially, the resulting calibration curves were indistinguishable whether in the absence or presence of (any kind of) internal standards. Herein, the calibration curves of trifluoroacetic acid (C2) under numerous types of interference were systematically examined, including other PFCAs, tethered fatty acids, untethered chemicals/ions, or environmental/biological matrices. The environmental concentrations of C2 were much higher than other PFCAs due to its diverse sources[29], with considerable rising rates recently and underestimated adverse health effects[43]. However, the quantitative analysis of C2 was arduous with a detection limit (LOD) among 10−500 ng L$^{-1}$ (Supplementary Table 6[28,29,44–48]). Our results showed that C4, C5 and C6 caused tiny deviation less than 1% on interval time (Fig. 3e), while the tethered short-chain fatty acids were easily distinguished by $\tau_{on}$, $I_\sigma$, $H_{peak}$ or other signal features (Supplementary Fig. 40), and a broad range of untethered common environmental and biological interferences, such as serum, tap water, urea, indole, ciprofloxacin, tris(2-carboxyethyl)phosphine (TCEP), lead (Pb$^{2+}$) and

chromium (Cr$^{2+}$) ions, caused insignificant change on baseline current or the major signal features of C2 (e.g., blockade and dwell time) even when the concentrations of those interferences were 20 times greater (Fig. 3f and Supplementary Figs. 41–42). Similar observations were also found in the quantification of C3 and C4 (Supplementary Fig. 43), highlighting the generality of anti-interference capability for nanopore based single-molecule sensing. Moreover, the LOD of C2 was further pushed down to 0.5 nM or 57 ng L$^{-1}$ (Fig. 3g), comparable to the state-of-the-art[28,29], by adopting an elevated voltage of -80 mV and a temperature of 30 °C to enhance driving force.

## Discussion

In short, this work developed a machine learning assisted, probe-directed nanopore single-molecule sensor towards the standard-free identification and quantification of multiple PFCAs simultaneously.

The standard-free identification was done in two steps, i.e., standard-free prediction of signal feature for unknown PFCAs based on the linear volume-blockade relationship, followed by high-resolution identification through multiple-feature classification with in-house feature library. The standard-free quantification was fulfilled by a "one-calibration curve-fit-all" strategy, or in other words, universal captures rates for various PFCAs. Both the standard-free identification and quantification of PFCAs were, in fact, determined by the double-barriers of probe-pore interactions between $R_6$ and aerolysin: the entry one could clearly impact the capture rates and calibration curves of PFCAs, while the exit one was essential to form the volume-current relation. Most importantly, it was possible to tune the strength of these two energy barriers independently, offering an opportunity to extend the standard-free quantification to medium- or long-chain PFCAs and to further lower down their LOD, without comprising the high-resolution standard-free identification capability.

As demonstrated above, WT aerolysin nanopore enabled a universal capture rate and standard-free quantification of C2/C3/C4, but when the chain lengths of PFCAs further increased from C5 to C8, their respective interval time under the same concentration was prolonged (Supplementary Fig. 33). To solve this issue, a mutated aerolysin nanopore, R282A, was designed to reduce entry barrier. All-atom molecular dynamics simulation displayed that the replacement of positive-charge arginine with short-chain hydrophobic alanine could effectively switch the electrostatic potential distribution around the cis entry of aerolysin nanopore (Figs. 4a, b), turning the repulsion force against $R_6$ probe into attraction. The steered molecular dynamics (SMD) analysis, i.e., to pull the PFCA-$R_6$ conjugate through aerolysin nanopore at a constant velocity, further revealed that the force required for motion (energy barrier) was comparable for C0/C2/C4 at the entry of WT aerolysin, but significantly higher for C6 (Fig. 4c), in line with experimental observation. In contrast, the R282A mutation exhibited a consistently lower entry barrier for C6 and C0/C2/C4 than the exit one and WT aerolysin (Fig. 4d). As a result, the voltage-driven capture pattern of C5 and C6 with R282A was akin to C2/C3/C4 and $R_6$ probe (Fig. 4e), expanding the coverage of standard-free quantification to almost 50% of PFCAs. Meanwhile, the LOD of C2 by R282A nanopore went down to 11.4 ng L$^{-1}$ or 0.1 nM at −70 mV and 30 °C (Fig. 4f), among the best of all existing techniques. Similar enhancement was observed in the LOD of C4 by R282A as well (Supplementary Fig. 44), validating the generality of single-molecule quantification for PFCAs.

Strikingly, it was also noted that the volume-blockade relations of WT aerolysin nanopore and the R282A mutant always followed a linear pattern and shared an identical slope of 0.023% Å$^{-3}$, no matter at mild or elevated voltages and temperatures (Fig. 4g and Supplementary Fig. 45). Besides, the identification accuracy by R282A at 30 °C/−70 mV remained close to 100% (Supplementary Fig. 46). This was because the positive-charge $R_6$ probe was able to drive various PFCAs to the same position of AeL nanopore lumen (Supplementary Fig. 47), due to the

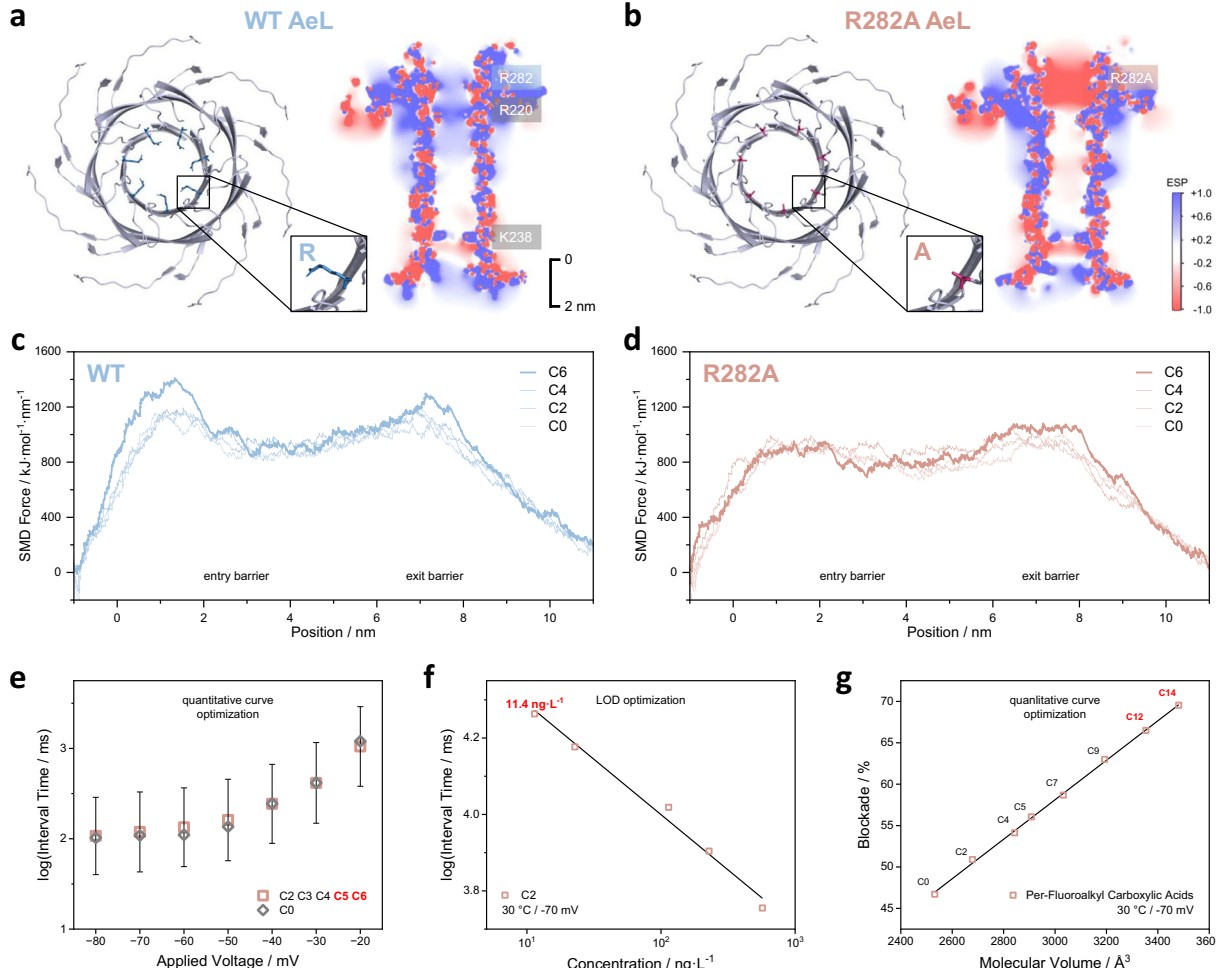

**Fig. 4 | Optimizing the double-barriers of probe-pore interaction towards directed evolution of standard-free single-molecule sensing of PFCAs. a, b** Top-down views of the cis entry for WT and R282A aerolysin nanopores, highlighting the structures of arginine or alanine residue at the 282 position, and the respective electrostatic potential distributions estimated at 4 M KCl, 20 °C and −50 mV. **c, d** SMD simulations for C0/C2/C4/C6-$R_6$ from cis entry to trans exit through WT and R282A aerolysin nanopores. **e** Voltage-dependent capture rates of C0/C2/C3/ C4/C5/C6-$R_6$ by R282A under the same concentration. The error bars were the average standard deviations obtained from the histograms of interval times of C2 to C6. **f** Calibration curve of C2 (linear fit, $R^2 = 0.9872$), measured with R282A at 30 °C and −70 mV. The concentration of C2 varied from 0.1 to 5 nM. **g** The linear volume-blockade relation measured with R282A at 30 °C and −70 mV (linear fit, $R^2 = 0.9989$), covering long-chain PFCAs up to C14.

combined effect of electrophoretic force on $R_6$ probe and the electrostatic interactions between $R_6$ and K238/K242 residues at the trans exit of AeL nanopore. This phenomenon strongly supported the above hypothesis that the standard-free quantification and identification activities could be independently improved as they were separately controlled by the entry and exit barrier of probe-pore interactions. Since the magnitude of blockade was smaller by R282A at −70 mV/ 30 °C, as a result of its higher open-pore current and greater driving force, the linear volume-blockade relationship was extended to C14, covering over 97% of PFCAs.

Lastly, a heavily engineered aerolysin nanopore, R282S/E216S/ R220S/E222S/A260F (4S1F), was computationally analysed to explore the feasibility of standard-free quantification of all PFCAs via single-molecule sensing. All four charged amino acids along the cis entry of aerolysin nanopore were substituted by a short-chain uncharged serine to reduce the entry barrier of PFCA-$R_6$ complex, while an additional cation-π interaction between AeL and $R_6$ probe was introduced at the trans exit to prolong the residence time of PFCAs in pore lumen. It was shown in SMD analysis that the entry barrier of C6 by 4S1F was 30% lower than R282A or 60% lower than WT

(Supplementary Fig. 48b), while the magnitude of exit barrier was maintained and its position was shifted towards the cis side. The lower entry barrier of 4S1F, or more precisely, greater capture rates of PFCAs, led to an identical distribution of energy barrier for C6/C8/ C10/C12/C14 (Supplementary Fig. 48c), indicating that this rationally designed mutant AeL nanopore would enable a universal capture rate or calibration curve for all PFCAs.

To summarize, our findings confirmed the feasibility to design suitable combinations of probe and nanopore towards single-molecule standard-free quantification of all PFCAs or PFAS, with the aid of machine learning. Based on the analysis of current predictions for all available PFCA isomers, Supplementary Fig. 49, it was seen clearly that blockade only would be difficult to reach both high resolution and wide coverage. Hence, a multi-feature classification approach was developed in this work, which had been experimentally proven capable of resolving more than 50 PFCAs, including 8 classes of 30 aliphatic or aromatic PFCA isomers[49–52] (Supplementary Figs. 50–51). After precisely engineering the entry barrier of AeL nanopore, it was shown experimentally and/or computationally that standard-free probe-directed nanopore single-molecule sensor was

able to identify and quantify multiple PFCAs (between C2 and C14) simultaneously under real environmental conditions.

Regarding the limitations of the current method, it should be noted that the volume-dependent structural assignment principle may encounter difficulties in standard-free identification, especially when dealing with unknown PFCA analogues or isomers of indistinguishable volumes (i.e., those yet to be included in the feature library). In this case, the current Supplementary Method 2 would still require a certain amount of prior knowledge, e.g., on the relative abundance of unknown PFCAs in real samples, so as to avoid the use of high-purity PFCA reference materials for the construction of single-molecule signal feature library. Without these, the insufficient blockade resolution of WT AeL nanopore could not unambiguously identify structurally diverse PFCA isomers in the absence of standards.

Looking forward, two research directions may mitigate the above limitations. One is to design suitable aerolysin mutants that could prolong the dwell times of PFCAs inside nanopore, narrow the distribution of blockades, reduce their prediction or identification errors, and eventually enhance the resolution for PFCA isomers of similar volumes. The other is to computationally predict additional nanopore signal features, leveraging the pre-established and fully resolved feature database, as well as the precisely controlled probe-nanopore interactions. Advances in deep learning, MD simulations, or generative modeling will support the formation of a more broadly structure-feature relationship. These potential improvements, while beyond the scope of the present work, illustrate possible paths for extending the utility of single-molecule sensing in PFAS analysis.

## Methods

### Reagents and chemicals
Potassium chloride (KCl, ≥99.9%), tris(hydroxymethyl)aminomethane (Tris, 99%), ethylenediaminetetraacetic acid (EDTA, ≥99.0%), octane (≥99%), n-caprylic acid (99%), and chromium(II) chloride (99.99%) were purchased from Titan Scientific (Shanghai, China). The electrolyte adopted in this detection system was the aqueous solution of 4 M KCl, 10 mM Tris, and 1 mM EDTA. 1,2-diphytanoyl-sn-glycero-3-phosphocholine (DPhPC, powder, ≥99%) was purchased from Avanti Polar Lipids (Shanghai, China). DPhPC was dissolved in octane and sealed dryly at -20°C. Proaerolysin was purchased from GenScript Biotech (Nanjing, China), and trypsin agarose (25UN) was purchased from Sigma-Aldrich (Shanghai, China). Conjugates of perfluorocarboxylic acids and peptide leaders were produced by ChinaPeptides (Suzhou, China). Tris(2-carboxyethyl)phosphine hydrochloride (TCEP, ≥98%) was purchased from Shanghai Dibai Chemical Technology (Shanghai, China). Ciprofloxacin (98%) was purchased from Macklin Inc. (Shanghai, China). Urea (99%), indole (98%), and lead chloride (99.99%) were purchased from Meryer (Shanghai, China). Fetal bovine serum was purchased from Hangzhou Gaosheng Biotech (Hangzhou, China).

### Single-molecule sensing of per- and polyfluoroalkyl carboxylic acids
Proaerolysin was activated with immobilized trypsin at 4 °C for 10 h to form monomeric aerolysin, which was involved in the oligomerization for aerolysin nanopores. After activation, proaerolysin was separated from trypsin at 10,000 RPM and stored at −20 °C for further use. The above electrolyte was added into the detection chamber, and the temperature was set as 20 °C constantly, except other stated. The DPhPC lipid bilayer membrane supporting the nanopore was formed across the 100 μm microcavities of MECA 4 Recording Chips (Nanion Technologies, Germany). Activated aerolysin was added into the cis chamber and formed a single nanopore on the DPhPC lipid bilayer membrane under the applied bias of +200 mV. The cis side of the detection chamber was seen as virtual ground whose potential was 0 mV and the voltage was applied on the trans side. If a single aerolysin nanopore was formed, open pore current would go to −100 pA as the applied voltage was set to −50 mV. PFCA-$R_6$ conjugates were added into the cis chamber and translocated to the trans side under negative voltages. Sampling rate and bandwidth were set as 20 and 10 kHz, respectively.

### Data acquisition and analysis
Electrical signals were recorded with an Orbit mini portable patch clamp amplifier (Nanion Technologies GmbH). Frequency-modulated signals were processed in MATLAB 2021b, via wavelet denoising or lowpass filter at 100, 200, 500, and 800 Hz. Machine learning models were trained, validated, and predicted in MATLAB as well. Features of raw and frequency-modulated signals were extracted in Python 3.11, based on the pre-identified events from Clampfit 10.4. Feature normalization was done in Python. Graphs were drawn in OriginPro 2024, MATLAB, or Python.

### Molecular volume simulation
Molecular volume simulations were conducted in GROMACS 2018.8. Three-dimensional structures of the $R_6$ probe and PFCA-R conjugates were generated in Chem3D 20.0. Energy minimization of these 3D structures was done with MM2 method and saved as pdb files. Structure topology files were generated with Ligand Reader and Modeler on CHARMM-GUI.

Molecules were simulated, and their volumes were calculated in an aqueous solution system. The aqueous solution system, including electrolyte, electric field, temperature, and pressure, was modeled and stabilized in advance. A $7 \times 7 \times 7$ nm³ sized water box was built through Solution Builder on CHARMM-GUI, with a KCl concentration of 4 M and temperature at 293 K. To eliminate the redundant steric hindrance and minimize the energy of this water box, 5000 iterations were done with the steepest descent algorithm. A 125 ps long equilibrium simulation was carried out to maintain constant temperature and reasonable distribution of total compounds. Based on this finished product, a 5 ns long simulation with a time step of 2 fs was carried out using Nose–Hoover method for temperature control and Parrinello–Rahman method for pressure control. Electrostatic interaction was calculated with Particle mesh Ewald (PME) method. Cut-off radius referring to calculations of Coulomb interaction, electrostatic interaction, and Van der Waals interaction was set as 12 Å. Calculations of hydrogen bonds were constrained with LINCS algorithm.

The last frame of the simulation was adopted as the water box for later molecular dynamic simulations. Afterwards, each structure under simulated was inserted into this water box with gmx insert-molecule, replacing water molecules, potassium ions, and chloride ions randomly. The whole system was controlled at 293 K with V-rescale method of temperature control. After 10,000 iterations with the steepest descent algorithm, energy minimization of this system was achieved. As a final step, there was a 1 ns long NPT equilibrium. Molecular volumes of all kinds of conjugates were calculated in a 50 ns long simulation, with time step of 2 fs. In purpose of reducing the impact of $R_6$ probe and highlighting real volume differences between different PFCA molecules, X-R conjugates (where X represented different PFCA targets) were designed and simulated first, putting the volume of X at a dominant position while taking the junction between the target molecule and the probe into consideration. The volume difference between C4-R and C4-$R_6$ was set as an internal standard to speculate all of the molecular volumes of X-$R_6$.

### Steered molecular dynamics simulation
To investigate the transmembrane dynamics of PFCAs tethered by the $R_6$ probe, we performed SMD simulations following the protocol below:

The wild-type aerolysin nanopore structure (PDB ID: 9FM6) was retrieved from the RCSB PDB database. A segment spanning residues 190–315 was extracted for subsequent mutagenesis and simulations.

Site-directed mutations were introduced into this truncated wild-type sequence to generate R282A and 4S1F (R282S/E216S/R220S/E222S/A260F) mutant nanopores. Since all nanopore variants followed identical simulation procedures, we described the methodology using the wild-type as an example.

The wild-type PDB structure was processed through the CHARMM-GUI web server for system construction. The initial simulation box was set to $10 \times 10 \times 15$ nm$^3$, solvated with 4 M KCl electrolyte solution. Simulations were conducted at 293.15 K using the CHARMM36m forcefield. System equilibration involved 0.125 ns in the NVT ensemble, followed by 1.5 ns in the NPT ensemble. Production simulations were subsequently carried out for 50 ns under NPT conditions to obtain fully equilibrated nanopore systems.

For the ligand molecule, C14-R$_6$ was initially sketched in Chem-Draw 20.0, and the structure was submitted to CHARMM-GUI's Ligand Reader & Modeler for parameterization using the General Force Field (GenFF). The resulting PFCAs structure was then positioned at the nanopore entrance using VMD 1.9.3. Water and ions within 2 Å of the ligand were removed from the system. The combined PFCAs-nanopore system underwent 5000 steps of energy minimization followed by 1.6 ns of SMD simulation. During SMD simulations, an external electric field of −50 mV was applied, with a pulling velocity of 0.01 nm/ps. The reference group was set to residue K230, while flexible positional restraints were applied to the C2 atoms of DPhPC and α-carbon atoms of the nanopore in the xy-plane. From the SMD trajectory, we extracted frame 2300 where the C14-R$_6$ molecule was fully extended. This frame was used as the starting structure for subsequent simulations, where additional flexible restraints were applied to the α-carbon atoms of the ligand.

For other PFCA molecules with varying chain lengths, structural modifications were performed in PyMol 2.6.2 by editing the straightened C14-R$_6$ structure through atom deletion. These modified molecules were then simulated following the identical protocol described above to obtain their respective SMD trajectories.

## Data availability
The data supporting the findings of the study are included in the main text and supplementary information files. Raw data can be obtained from the corresponding author on request. Source data are provided with this paper and available at https://doi.org/10.5281/zenodo.18595472, and https://doi.org/10.5281/zenodo.18611803. The structure of wild-type aerolysin nanopore for molecular dynamics simulation was retrieved from RCSB Protein Data Bank with the accession code: 9FM6. Source data are provided with this paper.

## Code availability
The custom MATLAB and Python scripts are available at https://doi.org/10.5281/zenodo.18595472.

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

## Acknowledgements

This study was supported by the National Key R&D Program of China (2023YFC3008803, to K. Q.), the National Natural Science Foundation of China (21972041 and 22006037, to K. Q.), and the Natural Science Foundation of Shanghai Municipality (23ZR1416300, to K. Q.).

## Author contributions

#These authors contributed equally. K.Q. conceived the project. J.Z., H.L., M.-Y.C., P.L., and X.X. performed the measurements. H.L., J.Z., and S.T. designed the machine learning algorithms. W.T., X.Z., Z.Y., and D.L. performed the MD simulation. K.Q., J.Z., and H.L. wrote the paper.

## Competing interests

The authors declare no competing interests.
