## [Transparent Peer Review file · Nature Communications]

Machine learning assisted single-molecule sensing towards standard-free quantification of per- and polyfluoroalkyl carboxylic acids

Corresponding Author: Dr Kaipei Qiu

Version 0:

Reviewer comments:

Reviewer #1

(Remarks to the Author)

Zou et al. describe one potential method to quantify PFCAs without standards using nanopore and machine learning. However, I do not get anyone successful case, in which this method can accurately and simultaneously quantify the levels of multiple PFCAs in various environmental matrices. The author is more demonstrating that this method may achieve standard free quantification. Certainly, I do not believe that this method can achieve standard free quantification based on the result from authors. Even, I think that all of work implied that PFCA standards are indispensable in this method, likely current method for PFCA. Except for trifluoroacetic acid, they do not provide any data to show the simultaneous quantification of other 12 PFCAs. Meanwhile, they do not discuss the influence from any interference in the environmental matrices. In addition, most techniques of nanopore and machine learning have not made significant advance. In summary, this manuscript is not appropriate for publication in Nature Communications, and it may warrant publication in an environmental analytical chemistry journal. My concerns are as following:

1. The authors claims that this solution can quantify PFCAs. However, they do not provide any case to quantify PFCAs in any environmental matrices, simultaneously compare with the performance of current solution, for example the HPLC-MS/MS method referred by authors. They spent a lot of text explaining the qualitative accuracy of this solution. But the quantitative accuracy is also important part for any methods. The only information about quantitative methods is the relationship between the logarithm of C2-PFCAs concentration and the logarithm of its interval time in Figure 3B. C2-PFCA is the most easily quantifiable PFCAs due to its lower molecular weight and a small amount of chemical structural analogues. With the growth of the carbon chain for PFCAs, the number of isomers and structural analogues will sharply increase. On the other hand, they give the best LOD for C2-PFCA is under -80mV and 30°C, not -50mV and 20°C. This implies that all of linear correlation between current blockades and the volumes of PFCA and machine learning model will be retested and reestablished. That is to say, there is no match between optimizing qualitative methods and optimizing quantitative methods. More importantly, this relationship between the logarithm of C2-PFCAs concentration and the logarithm of its interval time was established by PFCA standards. For PFCAs without commercial standards, we cannot establish corresponding relationship. This situation is exactly the same as current method, such as the HPLC-MS/MS method mentioned by the author. Thus, PFCA standards are indispensable in this method. In addition, the interval time, the factor used to quantify PFCAs, fluctuates greatly based on Figure S1. The authors do not offer any data on the uncertainty of quantification due to the measurement of interval time. Therefore, I believe that this method can not accurately and simultaneously quantify the levels of multiple PFCAs in various environmental matrices without standards.
2. Except for the quantitative methods, the qualitative accuracy is also deeply depended on standards. All of training data for machine learning model are acquired using PFCA standards. That is to say, if there is no commercial standard for PFCA, we cannot obtain their electrochemical signals for nanopore sensing, and therefore cannot use them for model training to identify the corresponding PFCA.
3. The linear correlation between current blockades and the volumes of PFCA simulated by molecular dynamics is the only potential way to identify PFCA without standards, but not quantify PFCA. But based on the result of author in Figure 1d, the identification of PFCA only using current blockades is not good, due to the overlap of current blockades among different PFCA, such as C5, C6, and FTA. In real scenarios, this overlap is more severe. This overlap is partly due to the presence of

more PFCA in practical environments, including linear, branched, cyclic isomers and heteroatom substituted, unsaturated analogues. On the other hand, it comes from structurally similar non-PFCAs, including oligomers, surfactants, and fatty acids. The authors also indicate that “the use of current blockade alone was unlikely to fully resolve the total 13 PFCA”

4. Although the authors firstly establish a linear correlation between current blockades and the volumes of PFCA simulated by molecular dynamics, this is not first discovery on the linear correlation between current blockades and the volumes of molecules. I do not believe that this is one obvious advance. Meanwhile, compared with current technology, the nanopore sensing system including aerolysin and oligo-arginine leader is known, and it also cannot make one significant advance. As the author stated, “Future studies towards the measurements of all the 338 PFCA isomers by SMS would require the rationale engineering of nanopore interface in the context of multiple feature classification.”, which is the research I am looking forward to seeing.

(Remarks on code availability)

Reviewer #2

(Remarks to the Author)

Key results

- The author have a sensor that they claim can “quantify a mixture of PFCAs with references.” They report the ability to predict molecular volumes from a linear relationship between current blockade and PFCA molecular volume.

Validity

- the rationale for the work is based on perceived limitations and technical challenges associated with current approaches for estimating concentrations of PFAS in humans and the environment.

Significance

- Quite a bit of the cited literature focuses on determinations of PFAS in environmental media, but the submitted manuscript does not attempt to demonstrate the technology in media of any complexity (e.g., electrolyte solution) and thus does not attempt to demonstrate the effectiveness of the technique in a system of any complexity. For example, natural waters contain organic matter, other trace organic and inorganic species. The suggestion in the manuscript is that this approach could replace current technology. The current manuscript does not bridge this gap. Thus, the significance of the technology is unclear.

- Per- and polyfluoroalkyl substances (PFAS) are a broad diverse set of molecules. The conventional definition of PFCAs is perfluoroalkyl carboxylates. The authors define PFACs as per- and polyfluoroalkyl carboxylic acid. Thus, the authors do not appear to be using conventional nomenclature and for this reason their training set is unclear and they are not listed in any of the accompanying documents that I could find.

Data and methodology, Analytical approach

- Since I was unimpressed by the potential significance of the paper, I did not access the methodology.

Suggested improvements

- An important step would be to attempt to demonstrate the technology in something other than pure electrolyte solutions and the chemical training set for the measurements should be revealed

References

- The references are fine but illustrate that the interest in these compounds centers on our ability to measure them in complex abiotic and biological systems.

Your expertise

- I have expertise in PFAS measurements and physical properties, but not machine learning.

(Remarks on code availability)

Reviewer #3

(Remarks to the Author)

The manuscript with ID NCOMMS-24-39141-T by Zuo and colleagues reports an interesting study on the development of a machine learning-assisted method to identify and quantify per- and polyfluoroalkyl carboxylic acids (PFCA—a subfamily of PFAS) using a single-pore electrochemical system. The proposed method can quantify and identify PFCAs without the need of reference standards based on the unique electrochemical fingerprint of these molecules when they flow across the nanopore. The study is somehow comprehensive in the generation of a broad database of PFCA fingerprints by assisted machine learning approach. However, I have serious reservations about the actual utility of the proposed scheme for real-life scenarios. My main points of criticism are summarised as follows:

1 – The study uses polycationic peptide probes to control the location of tethered PFCA when these flow through the

nanopore and induce the current blockade. This might be suitable for a laboratory study, where target PFCA is in its pure form. However, real-life PFCAs combine with other molecules in the environment. This alteration would introduce considerable errors in the identification.

2 – Another important aspect of is the use of highly concentrated electrolytes to control the surface charge. One of the biggest issues in the identification of PFCAs (and more broadly PFAS) is the interferences introduced by different water matrices with varying composition and ionic strength.

While I appreciate the technical merit of the study (I think that the approach is innovative and worth exploring), I do question its practical applicability. The use of standards is still a necessity to elucidate the effect of electrolyte and composition of PFAS in real-life water samples.

(Remarks on code availability)

The code is usable by the community

Reviewer #4

(Remarks to the Author)

Environmental contamination by perfluoroalkyl and polyfluoroalkyl substances (PFAS) is a notable global concern. Current methods for detecting PFAS typically require standard samples, which are limited in availability. Therefore, methods that do not rely on standard samples are highly desirable. The authors proposed using wild-type aerolysin nanopores in a 4-M KCl solution for detecting perfluorocarboxylic acids (PFCAs) without standard samples. They reported a strong linear relationship between molecular volume and current and achieved almost 100% accuracy in identifying PFCAs using the bag tree classification model. They also demonstrated that PFCAs can be quantified even in the presence of interference.

1. Although the introduction aims to quantify PFAS without relying on standard samples, the paper does not fully address this issue. The authors mention that there are over 14,000 PFAS chemical species but only about 120 standard samples. Their study focuses on nanopore measurements of PFCAs (C2–C9) and some of their substituents. Since most PFAS molecules lack carboxyl groups, even if the method accurately predicts PFCAs with carboxyl groups, its applicability to PFAS without carboxyl groups is uncertain. This paper does not demonstrate a method that can be applied to molecules without carboxyl groups, leaving the initial problem unsolved.

2. The authors achieved high classification accuracy by constructing a machine learning model with multiple features. However, supervised learning typically requires standard samples. To obtain measurements without relying on standard samples, it may be more appropriate to develop a regression model that predicts the properties of molecules not included in the training data. For instance, a regression model could estimate molecular volume or carbon number using the same feature set. However, even with such analysis, eliminating the need for standard PFAS samples may still be not feasible.

3. The study uses the linearity between excluded volume and current to support a quantification method without standard samples. Although linear chains of PFCAs have been successfully measured and predicted, branched-chain PFCAs also exist. For example, can PFCAs having a trifluoromethyl group (–CF₃) be classified using the same approach?

4. The model was validated only for PFCA molecules up to C9. Can the model be applied to longer molecules? For instance, C14 is regulated by the European Union; can the model be extrapolated to accommodate C14?

5. The authors attribute the observed linearity to polycationic peptide probes and concentrated electrolytes. However, could this linearity be a molecular characteristic? Is similar linearity observed in non-fluorinated molecules?

6. The environment may contain fatty acids that could interfere with the measurements. Is the method still effective if fatty acids with similar volumes are present?

7. A reproducible description of the analysis method or the source code should be made publicly available, including details of the mother wavelet used. Because nanopores are laboratory-fabricated rather than commercially produced, publicly available measurement data can enhance the reliability of the obtained results, particularly in research that integrates nanopores with artificial intelligence.

(Remarks on code availability)

Reviewer #5

(Remarks to the Author)

(Remarks on code availability)

Version 1:

Reviewer comments:

Reviewer #1

(Remarks to the Author)

I am truly grateful for the author's efforts in responding to my comments. Many of my comments have also received responses. However, the evidence that supporting methods can standard-freely quantify multiple PFAS under real environmental conditions is still insufficient.

1. I am concerned about the generality of the standard-free quantification. In standard free quantification, the author used ultra short chain and short chain PFCAs (C2/C3/C4) as demonstrations, and they also provided the voltage-dependent capture pattern of C0, C2, C3 and C4. But how about the capture pattern along with the increasing of their concentration? The author also give reason for this, "as the electrostatic driving force on polycationic peptide was much greater than the entry barrier into AeL nanopore for ultrashort- or short-chain PFCA." But how about the capture pattern of medium and long chain PFCAs? These results will impact whether one concentration-interval time response curve fit all different kinds of PFCA.

2. I am concerned about the accuracy of the standard-free quantification in real environmental samples. Although the author has tested C2 in Figure 3c and S37, but how about medium and long chain PFCAs? In our previous comments, we also mentioned that the isomers of C2 are very limited and face little interference from environmental analogues. As the carbon chain grows, this problem theoretically becomes more and more serious. In Figure S37, the author tested all of untethered inferences for C2, how about tethered inferences, C3, C4, or haloacetic acid (difluoroacetic acid)?

3. I do not agree with the response of using different experimental conditions for qualitative and quantitative analysis, because standard free quantification requires standard free qualitative analysis first. Although the relationship between current and molecular volume or machine learning models can be reestablished, the slope of the linear relationship under different experimental conditions affects the resolution of qualitative analysis. Unless the author provides evidence to prove that the resolution of qualitative analysis is similar under the two conditions, I do not agree that quantitative analysis can be superior to LC-MS/MS and GC-MS. In addition, the two articles are also papers from 14 years ago.

4. Although under the testing conditions of laboratory standards, the detection limit of C2 is lower than existing methods (HPLC-MS/MS and GC-MS). However, in the current HPLC-MS/MS method, the detection limit of C2 is the highest compared to other PFASs (0.1 ng/L). Based on the existing results, we cannot speculate that the detection limit of other PFASs using single-molecule sensing will be better than the current HPLC-MS/MS method.

5. The author provides a roadmap for constructing a machine learning model for standard-free identification. But the key to this approach lies in the ability of unsupervised clustering analysis to distinguish different PFCAs (workflow S2), which directly affects how much blockage or what type of PFCA for one single molecule signal is labeled as, and thus affects the training of machine learning models used for standard free qualitative analysis. Moreover, the author did not introduce this unsupervised clustering analysis, nor did they discuss its discriminative ability. I assume that if two groups of single-molecule signals with the same blockade are discovered through this unsupervised clustering analysis, how should we distinguish which PFCA they correspond to? This is very likely. According to Figure S41a provided by the author, we can observe that PFCAs with different molecular weights may have similar molecular volumes, although this figure mainly reflects the molecular volume differences of isomers of PFCAs with the same molecular weight. The reason why the machine learning model for standard-free identification in this article is so good, I speculate, is because the selected PFCAs have significant differences in blockade and can be completely separated by the unsupervised clustering analysis. Based on this, I am concerned that PFCAs or even non PFCAs that did not participate in the training of model may affect the accuracy of standard-free identification. And this concern can be eliminated by using PFCA standards.

In summary, based on the current data of the author, I believe that the author can achieve the qualitative and quantitative analysis of C2 without standards in real environmental samples, and this process also requires the use of internal standards to reduce measurement errors, which is very similar to LC-MS/MS and GC-MS. However, if this is the case, the machine learning model used for standard free identification is not necessary, and only the linear relationship between molecular volume and blockade is needed to achieve the qualitative analysis of C2 without standard samples. These results still do not achieve significant progress.

(Remarks on code availability)

Reviewer #3

(Remarks to the Author)

The Authors have made sincere efforts to address all my criticism. This includes an extensive piece of additional work aiming at addressing my questions and commentary. Although I believe that there remain critical aspects to be fully addressed in order to demonstrate the practical applicability of the proposed sensing system to detect PFCAs in a range of real-life scenarios and validate the ML component of the study, it is my opinion that the study has fundamental value and brings a new perspective into a complex issue in analytical chemistry. The Authors have put their system under a considerable scrutiny, testing and benchmarking the system's performance with the presence of interfering molecules. I still believe that a broader range of interference compounds should be explored in order to fully validate the system. However, this is a practical aspect that should not remove merit to the fundamental contribution of the study, which is indeed substantial. For these reasons and based on my own personal assessment, I would recommend the study for publication in Nature Communications.

(Remarks on code availability)

The code can be used by the community and reproduced.

Reviewer #4

(Remarks to the Author)

The authors fully improved the manuscript and addressed my questions satisfactorily. I think that the manuscript is ready for publication in Nature Communications.

(Remarks on code availability)

REDAME is described in a straightforward manner.

Raw data preprocessing was performed using the commercial software MATLAB.

There were no bugs in the Python code.

Reviewer #5

(Remarks to the Author)

(Remarks on code availability)

Version 2:

Reviewer comments:

Reviewer #1

(Remarks to the Author)

While the authors have demonstrated good quantitative performance of their method with real samples, I wish to highlight that the current methodological framework cannot achieve truly "standard-free identification" of PFAS isomers.

The results clearly show that blockade signal alone is insufficient for effective isomer differentiation:

1. "The classification accuracy by blockade only was only 61.8% for all seven isomers."

2. "The maximal peak separation for blockades was just 3.2% among 17 isomers (Figure S57d), leading to less than 50% classification accuracy... and an overall accuracy of 31.1%... using blockade only (Figure S58a)."

Consequently, when using Step d of Workflow 2 for PFCA structural identification, multiple possible PFAS matches will fall within the molecular volume prediction error range.

I acknowledge that supervised or unsupervised clustering algorithms can leverage "many other features of the single-molecule signals, other than blockade" to distinguish different PFAS. However, the relationship between these features and molecular structures remains unclear. This means:

- These features can only be measured using known PFAS standards.
- Unlike molecular volume (which can be predicted from structure), these features cannot be computationally predicted for novel/unidentified PFAS structures.

Therefore, PFAS identification still fundamentally relies on:

1. The availability of PFAS standards for measurement, OR
2. Pre-existing databases built using such standards.

Both approaches are limited by the number of PFAS standards available.

The current method primarily uses the predicted molecular volume database for final structural assignment. This creates a significant limitation: identification is confounded by PFAS sharing similar molecular volumes. The authors' own example distinguishing FA4 and C3 illustrates this:

- If FA4 were an unknown PFAS with a similar volume, the method could only determine it is not C3. Its specific identity would depend entirely on whether a matching structure exists in the predicted volume database.
- While the FA4/C3 case might be hypothetical, the potential for confusion is demonstrably real for the medium/long-chain PFAS isomers studied in the authors' previous work (e.g., the 17 H-substituted positional aromatic PFCA isomers or the seven typical aliphatic PFCA isomers). In those studies, the authors had standards and successfully used clustering to differentiate them. Crucially, if some of these were unknowns, Workflow 2 could not definitively assign their structures due to volume similarities.

Recommendation:

Given these limitations, the authors should refrain from emphasizing "standard-free identification" as a key capability of the current method. The dependence on standards (either directly or via pre-built databases) and the inherent limitation posed by similar molecular volumes should be explicitly stated in the manuscript's limitations section.

(Remarks on code availability)

Version 3:

Reviewer comments:

Reviewer #1

(Remarks to the Author)

The authors have done a good job addressing the revisions. The limitations of the method have been effectively supplemented, and claims regarding the "free-standard identification" have also been appropriately restricted to the pre-established database. I have no further comments.

(Remarks on code availability)

Rebuttal Letter

Reviewer #1 (Remarks to the Author):

Review Comments:

Zuo et al. describe one potential method to quantify PFCAs without standards using nanopore and machine learning. However, I do not get anyone successful case, in which this method can accurately and simultaneously quantify the levels of multiple PFCAs in various environmental matrices. The author is more demonstrating that this method may achieve standard free quantification. Certainly, I do not believe that this method can achieve standard free quantification based on the result from authors. Even, I think that all of work implied that PFCA standards are indispensable in this method, likely current method for PFCA. Except for trifluoroacetic acid, they do not provide any data to show the simultaneous quantification of other 12 PFCAs. Meanwhile, they do not discuss the influence from any interference in the environmental matrices. In addition, most techniques of nanopore and machine learning have not made significant advance. In summary, this manuscript is not appropriate for publication in Nature Communications, and it may warrant publication in an environmental analytical chemistry journal. My concerns are as following:

Author Response:

Thank you for the time and efforts to help improve the quality of our manuscript. With regard to the three major concerns you raised, i.e., “PFCA standards are indispensable for quantification”, “lack of demonstration for simultaneous quantification in complex matrix”, “no significant advance over existing nanopore and machine learning techniques”, a point-by-point response are provided:

- 1) The standard-free quantification can be achieved in three steps: a) standard-free prediction of the blockade for unknown PFCAs using the linear volume-current relation; b) standard-free identification of a wide range of PFCAs via the high-resolution frequency-modulated multiple feature classification; c) standard-free quantification of various PFCAs using the probe-directed capture. Notably, since all the features are extracted from individual single-molecule signals, it is facile to derive other related features by the standard-free prediction of blockade and unsupervised clustering (*Workflow S1-2*). Meanwhile, we have shown in the revision that the capture rates of different PFCAs could be decided solely by the polycationic peptide probes, enabling the universal single-molecule counting for absolute quantification. Putting these together, it is possible to avoid the use of extraction standard for recovery analysis in complex matrices, injection standard for chromatography/mass spectrometry in identification, and native standard for building calibration curve in quantification, which are all mandatory in target analysis;
- 2) Multiplexing in various environmental or biological interferences is actually a technically solved issue for single-molecule sensing, but as most reviewers raise their concern on it, we have added 60 quantitative measurements with a total of nine different interferences, including simultaneous quantifications of multiple species, to validate its feasibility. In the meantime, a detailed workflow has been added in Fig. 3a to describe how probe-directed single-molecule sensing is applied in real-world standard-free quantification of PFCAs. Briefly, polycationic probes are first incubated with analytes to tether the PFCA targets on peptide leaders, which is a common step in nanopore technology, and has been verified in

this work as well (Fig. S40). Untethered inferences have little effect on PFCA analysis, no matter whether it is membrane stability, identification or quantification performance (Fig. 3c, S36-37). Meanwhile, all those tethered PFCAs can be determined simultaneously via multi-feature classification (Fig. 3b, S32-35);

- 3) The strict linear correlation established in this work ($R^2 = 99.98\%$ for 13 PFCA) between the volume and current blockade of PFCA was higher than any other nanopore work, i.e., R^2 from 73.34% to 93.43% (Nat. Biotechnol. 2020, 38, 176; Nat. Commun. 2021, 12, 5795; ACS Nano 2023, 17, 13685; ACS Nano 2020, 14, 2296; ACS Nano 2022, 16, 7258). It was worth noting that the volume-current relationship must be strictly linear so that it could make the volume-exclusion effect the dominant factor and be applied for accurate prediction of all kinds of PFCA. More importantly, this work developed for the first time a frequency-modulated multiple feature classification method that could remarkably increase the number of available features and prioritize their rank for few-shot learning. A total of 43 features were designed in this work, while most of the single-molecule analysis only used 2-9 features (Nat. Methods 2024, 21, 92; Nat. Nanotechnol. 2022, 17, 976; Nano Lett. 2023, 23, 9437; Nano Lett. 2023, 23, 8620; J. Am. Chem. Soc. 2022, 144, 13717; Angew. Chem. 2022, 134, e202203769). Shortlisting the 21-feature model reduced the required test set by 7.6 folds, which indicated that a total of 20 signals in 10 minutes were enough for PFCA determination even at the interference of 100 times concentration.

1. The authors claims that this solution can quantify PFCAs. However, they do not provide any case to quantify PFCAs in any environmental matrices, simultaneously compare with the performance of current solution, for example the HPLC-MS/MS method referred by authors. They spent a lot of text explaining the qualitative accuracy of this solution. But the quantitative accuracy is also important part for any methods. The only information about quantitative methods is the relationship between the logarithm of C2-PFCAs concentration and the logarithm of its interval time in Figure 3B. C2-PFCA is the most easily quantifiable PFCAs due to its lower molecular weight and a small amount of chemical structural analogues. With the growth of the carbon chain for PFCAs, the number of isomers and structural analogues will sharply increase. On the other hand, they give the best LOD for C2-PFCA is under -80mV and 30°C, not -50mV and 20°C. This implies that all of linear correlation between current blockades and the volumes of PFCA and machine learning model will be retested and reestablished. That is to say, there is no match between optimizing qualitative methods and optimizing quantitative methods. More importantly, this relationship between the logarithm of C2-PFCAs concentration and the logarithm of its interval time was established by PFCA standards. For PFCAs without commercial standards, we cannot establish corresponding relationship. This situation is exactly the same as current method, such as the HPLC-MS/MS method mentioned by the author. Thus, PFCA standards are indispensable in this method. In addition, the interval time, the factor used to quantify PFCAs, fluctuates greatly based on Figure S1. The authors do not offer any data on the uncertainty of quantification due to the measurement of interval time. Therefore, I believe that this method can not accurately and simultaneously quantify the levels of multiple PFCAs in various environmental matrices without standards.

Thank you for the comments on the quantification performance of single-molecule sensing of PFCAs. A total of six issues have been raised, i.e., 1) simultaneous quantification ability of multiple

PFCAs; 2) quantification reliability in the presence of interferences from real-life environments; 3) quantification uncertainty due to the measurement of interval time; 4) feasibility towards standard-free quantification of PFCAs; 5) mismatch between the measurement conditions for identification and quantification; 6) comparison of single-molecule sensing of PFCAs with other existing methods. Detailed, point-by-point responses are given below. Briefly, in the revision, we have provided extra experimental results to solve *Issues 1-4*, and included necessary explanations to justify *Issues 5-6*.

1) Simultaneous quantification ability of multiple PFCAs

Since the simultaneous quantification of two or more analytes is a technically solved issue for single-molecule sensing (*Nature* **1999**, 398, 686; *Nat. Biotechnol.* **2000**, 17, 1005), we didn't discuss it in detail due to the limit of length for original submission. To illustrate its practicability, especially to general audience, we have added in the revision the step-by-step experimental and data analysis process for the simultaneous determination of multiple PFCAs (*Workflow S3*), with typical examples of quantifications of C2 in the presence of C4/C5/C6 (*Fig. 3b, S32-34*), as well as detection of 3H and 3Cl with C5 (*Fig. S35*) or C5 with C2 (*Workflow S3f*).

Workflow S3. Simultaneous quantification of mixed analytes.

Fig. 3 (b) Quantification of C2 in the presence of C4, C5 or C6 at 20°C and -50 mV. The concentration of C2 varied from 50 nM to 100 μM, while those of C4, C5 and C6 were kept at 6.7, 12.5 and 12.5 μM, respectively.

Fig. S35 Simultaneous quantification instances of (a) 3H and (b) 3Cl under the interference of C5-R₆.

2) Quantification reliability in the presence of interferences from real-life environments

To validate the reliability of PFCA single-molecule sensing in complex environmental matrices, we have performed 60 quantitative measurements with a broad range of common environmental or biological interferences, such as serum, tap water, urea, indole, ciprofloxacin, caprylic acid, TCEP, Pb^{2+} , and Cr^{2+} . In most cases, these interferences caused insignificant change on the baseline current, and thus the major signal features of C2/C0 (e.g., blockade and dwell time) remained constant even when the concentrations of those interferences were 20 times greater (Fig. S36-37). As a result, the interference-free quantification was readily achievable (Fig. 3c).

Fig. 3 (c) Interference-free quantification of C2 in the presence of serum or tap water at 20°C and -50 mV. The concentration of C2 varied from 500 nM to 100 μ M.

Fig. S36 (a) Raw current traces without and with interference - urea, Tris(2-carboxyethyl)phosphine (TCEP), Pb^{2+} , indole, caprylic acid. (b-d) $\Delta I/I_0$, τ_{on} and τ_{off} of C0 at different concentrations of environmental or biological interferences. The concentration of C0 is 4 μ M in urea and TCEP, and 2 μ M in Pb^{2+} , indole and caprylic acid.

Fig. S37 (a) Raw current traces without and with interference - serum, TCEP, Cr²⁺, Ciprofloxacin. (b-d) $\Delta I/I_0$, τ_{on} and τ_{off} of C2 at different concentrations of environmental or biological interferences. The concentration of C2 is 2 μ M.

3) Quantification uncertainty due to the measurement of interval time

It is correct that the specific interval time between two individual signals may fluctuate greatly, but the average interval time is able to remain constant as long as the statistically sufficiently signals are captured. To demonstrate this capability, we divided a full current trace into pieces of 2.5 s with a moving step of 0.5 s, and calculated the mean interval time in each piece (*Fig. S39*). It was found that although individual times varied between 30-48 ms, the mean values over this 480 s are almost identical at 36.7 ms. In other words, the quantification uncertainty of interval time could be largely reduced via oversampling.

Fig. S39 Interval time calculated at different analysis time.

4) Feasibility towards standard-free quantification of PFCAs

Thank you for the critical comment that the plotting of calibration curve for quantification may still need PFCA standards. To address this issue, we initiated a probe-directed capture strategy, and

provided a proof-of-concept demonstration that the capture rate of short-chain PFCAs was decided by polycationic probes, enabling the universal single-molecule counting for absolute quantification. It was first noticed that the voltage-dependent capture pattern of C2, C3 and C4 was almost identical to that of the R₆ probe (Fig. 3e), as the electrostatic driving force on polycationic peptide was much greater than the entry barrier into AeL nanopore for ultrashort- or short-chain PFCA. Besides, six different molar mixing ratios of C2/C3/C4 were measured (three repetitions for each combination) – by normalizing the number of signals for the fewest analyte to 1, the averaged numbers of signals for the other two were very close to 2 and 3, consistent with the concentration ratio (Fig. 3f), which confirmed that one concentration-interval time response curve could fit all different kinds of PFCA, thus potentially avoiding the use of standards for quantification.

Fig. 3 (e) Comparison of the voltage-dependent capture rates for C2, C3 and C4, as well as the R₆ probe. (f) Comparison of signal ratios for the mixture of C2/C3/C4 with molar ratios of 1:2:3, 1:3:2, 2:1:3, 2:3:1, 3:1:2, and 3:2:1, respectively. The number of signals for the fewest analyte was normalized to 1.

5) Mismatch between the measurement conditions for identification and quantification

Thank you for this critical comment. We would like to emphasize that the test conditions used for the evaluation of PFCA identification performance, i.e., -50 mV and 20°C, have already enabled the interference-free quantification in a wide concentration range of 50 nM to 100 μM for C2, which is satisfactory for proof-of-concept. To demonstrate the feasibility of further lowering down the limit of detection, two most facile approaches to enhance driving force were examined, i.e., at an elevated voltage of -80 mV and temperature of 30°C, and the resulting LOD was comparable to the state-of-the-art performance. It is true that the development condition for identification is different from the optimized condition for quantification at the moment, but such mismatch is well-accepted in single-molecule studies (*Nat. Nanotechnol.* **2011**, *6*, 668; *J. Am. Chem. Soc.* **2011**, *133*, 18312). Moreover, as long as the exit barrier of AeL is greater than the entry one, R₆ peptide probes are always able to drive the non-ionic PFCA targets to the identical position in nanopore, leading to the establishment of a linear volume-current relationship.

6) Comparison of single-molecule sensing of PFCAs with other existing methods:

In fact, we have already included in the original submission some comparison of quantification performance of single-molecule sensing with existing methods. For instance, the LOD of C2 under the optimal condition could reach 57 ng·L⁻¹ (Fig. 3d), which was comparable to the state-of-the-art performance (10-500 ng·L⁻¹) for LC-MS/MS or GC (Table S5). Furthermore, the anti-interference of single-molecule sensing is superb. Even in the presence of interference of 100 times concentration,

nearly 80% quantification reliability could be maintained, which was at least one order of magnitude better than ensemble analysis, i.e., mimicked by the multi-peak fit with 2000 Hz blockade (Fig. 2g).

Fig. 3 (d) Quantification of C2 at 30 ° C and -80 mV to lower the limit of detection. The concentration of C2 varied from 0.5 to 50 nM.

Table S6. The limit of detection of C2 in previous research.

Year	Journal	LOD	Methods	Ref
2024	Environ. Sci. Technol	13	GC-ECD	41
		110		
2024	Environ. Sci. Technol	500	LC-MS/MS	28
2023	Environ. Sci. Technol	27	IC-MS/MS	43
2023	Environ. Sci. Technol	35	HPLC-MS/MS	44
2023	Environ. Sci. Technol	172	HPLC-MS/MS	45
2023	Anal. Chem.	10	UPLC-MS/MS	27
2022	Environ. Sci. Technol	19.5	GC-ECD	46
		95.1		

Fig. 2 (g) The identification accuracy of FTA in presence of C5/C6 by ensemble analysis (mimicked by the multi-peak fitting using blockade only), the full 43-feature model, and the shortlisted 21-feature one. The concentration ratio of FTA to the C5/C6 interference ranged from 1:1 to 1:1000.

2. Except for the quantitative methods, the qualitative accuracy is also deeply depended on standards. All of training data for machine learning model are acquired using PFCA standards. That is to say, if there is no commercial standard for PFCA, we cannot obtain their electrochemical signals for nanopore sensing, and therefore cannot use them for model training to identify the corresponding PFCA.

Thank you for this insightful comment. However, the construction of feature library does NOT rely on PFCA standards. Since all the features are extracted from individual single-molecule signals, the standard-free prediction of blockade is able to derive other related features, regardless of using pure or mixed PFCAs, as long as complete discrimination is realized through high-resolution multi-feature classification or clustering. In fact, even when pure PFCA samples (not necessarily standards) are commercially available, they need to be measured with some other known PFCA, e.g., C3/C5/C6 in this work, to calibrate their blockade and reduce experimental errors (*Workflow S1*). With regard to unknown PFCAs (*Workflow S2*): a) the frequency-modulated multiple dimensional features of all current events were extracted, including the signals for calibration PFCA; b) unsupervised clustering was applied to separate and aggregate the current events of unknown PFCAs into certain groups; c) the calibrated blockade of each group was compared to the volume-current relationship (*Fig. S41*) for standard-free identification; and d) other multi-dimensional features of the identified PFCA were labelled and added into the signal database. Hence, the ultra-high resolution of the multiple-feature classification and clustering is crucial, and we have confirmed recently that PFCA isomers could be accurately identified with this approach (<https://link.cnki.net/urlid/11.2097.X.20240927.1122.004>).

Workflow S1. Conventional data acquisition, feature extraction and model training with known PFCA samples.

Workflow S2. Standard-free identification and feature extraction of unknown PFCA.

3. The linear correlation between current blockades and the volumes of PFCA simulated by molecular dynamics is the only potential way to identify PFCA without standards, but not quantify PFCA. But based on the result of author in Figure 1d, the identification of PFCA only using current blockades is not good, due to the overlap of current blockades among different PFCA, such as C5, C6, and FTA. In real scenarios, this overlap is more severe. This overlap is partly due to the presence of more PFCA in practical environments, including linear, branched, cyclic isomers and heteroatom substituted, unsaturated analogues. On the other hand, it comes from structurally similar non-PFCAs, including oligomers, surfactants, and fatty acids. The authors also indicate that “the use of current blockade alone was unlikely to fully resolve the total 13 PFCA”.

It is absolutely correct that the resolution of blockade is not sufficient for PFCA identification, and that’s why we have developed in this work the frequency-modulated multi-feature classification to enhance its resolution. As stated in the previous response, the standard-free prediction of blockade is able to derive other related features because all the features are obtained from an individual single-molecule signal, which is remarkably different from ensemble analysis.

To highlight the broad applicability of the established linear volume-blockade relationship, a library of current signatures has been built for all the 311 PFCA isomers with a molar mass less than 500 Da (*Fig. S41*), while its prediction accuracy is further proven by five distinct PFCAs containing methyl/hydroxyl side chains, double bonds or aromatic groups (*Fig. S6*). The ultra-high resolution of the proposed frequency-modulated multi-dimensional feature classification has been consistently observed in our recent experiments on tens of other PFCA isomers (unpublished) – if you and other reviewers consider that it is necessary to include more results, we are happy to share those data.

Moreover, a detailed workflow has been provided in *Fig. 3a* to better illustrate the application of standard-free single-molecule quantification of PFCAs in realistic scenarios. Briefly, polycationic probes are first incubated with analytes to tether the PFCA targets onto peptide leaders, which is a common pre-processing step in nanopore technology (<https://nanoporetech.com/products/prepare>), and has been verified in our work as well (*Fig. S40*). Untethered inference has negligible effect on PFCA analysis, no matter whether it is membrane stability, identification or quantification accuracy (*Fig. 3c, S36-37*); while all the tethered PFCAs are determined simultaneously through multi-feature classification (*Fig. 3b, S32-35*).

Fig. 3 (a) The generalized workflow for probe-directed single-molecule sensing of PFCA with the

assistance of machine learning. (b) Quantification of C2 in the presence of C4, C5 or C6 at 20 °C and -50 mV. The concentration of C2 varied from 50 nM to 100 μM, while those of C4, C5 and C6 were kept at 6.7, 12.5 and 12.5 μM, respectively. (c) Interference-free quantification of C2 in the presence of serum or tap water at 20 °C and -50 mV. The concentration of C2 varied from 500 nM to 100 μM.

Fig. S6 Experimentally measured current blockade and molecular volume of poly-fluoroalkyl carboxylic acids with other structures, along with the established volume-current relationship (grey line).

Fig. S40 Typical blockade and standard deviation signals for (a) the R₆ probes (C0) and (b) the incubation of R₆ probes with different concentrations of activated C2/C4/C6. The overall duration for activation and incubation is roughly 1 hour.

Fig. S41 (a) Calculated molecular volumes for PFCA isomers with a molar mass less than 500 Da. (b) Distribution of molecular volumes and blockades of PFCA isomers. (c) Percentage of fully discriminated PFCA isomers compared to the standard deviation of current blockades.

4. Although the authors firstly establish a linear correlation between current blockades and the volumes of PFCA simulated by molecular dynamics, this is not first discovery on the linear correlation between current blockades and the volumes of molecules. I do not believe that this is one obvious advance. Meanwhile, compared with current technology, the nanopore sensing system including aerolysin and oligo-arginine leader is known, and it also cannot make one significant advance. As the author stated, “Future studies towards the measurements of all the 338 PFCA isomers by SMS would require the rationale engineering of nanopore interface in the context of multiple feature classification.”, which is the research I am looking forward to seeing.

Thank you for the comments and suggestions. We would like to emphasize that the proposed single-molecule sensing of PFCAs was a digital quantification method (*Nature* **2024**, 628, 771; *Nat. Nanotechnol.* **2024**, 19, 800; *Nat. Commun.* **2023**, 14, 653), with three consecutive breakthroughs:

a) *standard-free prediction* of the blockade for unknown PFCAs using a strict linear volume-current relationship; b) *standard-free identification* of a wide range of PFCAs via the high-resolution multi-feature classification; c) *standard-free quantification* of multiple PFCAs in complex matrices using the probe-directed capture. Putting all these together, it is thus possible to avoid the use of extraction standards for recovery calculation in complex matrices, injection standards for LC/MS-MS or GC in identification, and native standards for building calibration curve in quantification, which are all mandatory in conventional target analysis. The feasibility of standard-free prediction, identification and quantification have been thoroughly examined in the revised manuscript, which hopefully can convince you this time.

Scheme 1. Comparison of standard-free single-molecule sensing of PFCAs against LC-MS/MS.

As mentioned above, the ultra-high resolution for complete discrimination of all PFCAs in the mixture is a prerequisite to standard-free quantification. To achieve this purpose, we have developed for the first time a frequency-modulated multi-feature classification method to significantly increase the number of available features and prioritize their rank for few-shot learning. The identification accuracy of PFCAs (test set) was *improved from 89.1% (1 feature), to 96.7% (8 features), and further to 99.9% (43 features)*, while the latest advances in single-molecule analysis of amino acids, riboses, alditols, saccharides, or benzenediols used 2-9 features only (see below for details), and 2 features were sufficient in most cases. Such a finding indicated that although the resolution of blockade in the linear range of volume-current relationship was 2 orders of magnitude lower than required to resolve all the PFCAs identified previously, the frequency-modulated multi-feature method was able to do so without modifying nanopore interface. More importantly, the shortlisted 21-feature model reduced the required test set by 7.6 folds, suggesting that *a total of 20 signals in 10 minutes were enough for PFCAs determination even at the interference of 100 times concentration*.

- 2 features (96.0% test) or 5 features (98.8% test) for 20 amino acids (*Nat. Methods* **2024**, 21, 92);
- 2 features (99.6% test) for 11 riboses (*Nat. Nanotechnol.* **2022**, 17, 976);
- 5 features (99.9% validation) for 10 riboses (*Nano Lett.* **2023**, 23, 9437);
- 5 features (99.4% validation) for 5 riboses (*Nano Lett.* **2023**, 23, 8620);
- 7 features (99.4% validation) for 13 alditols (*J. Am. Chem. Soc.* **2022**, 144, 13717);
- 9 features (95.9% test) for 9 saccharides (*Angew. Chem. Int. Ed.* **2022**, 61, e202203769);
- 5 features (99.9% validation) for 12 riboses (*ACS Nano* **2022**, 16, 21356);
- 2 features (98.2% validation) for 4 benzenediols (*ACS Nano* **2022**, 16, 6615).

What's more, as recognized by you and other reviewers, our work has realized for the first time a *strict* linear correlation (i.e., $R^2 = 99.98\%$ for 13 PFCAs) between the volume and current blockade of PFCAs, which is the other prerequisite to standard-free prediction and identification. In contrast,

recent progress in single-molecule analysis only reached R^2 of 73.34% to 93.43% based on the mass of amino acids, peptides, or proteins (see below for details), despite the volume-current relationship was much sounder theoretically (*Nat. Biotechnol.* **2020**, 38, 176). It is correct that similar strategies have been explored, but our performance is considerably improved because we figure out correctly the significance of using polycationic probes to control the position of tethered PFCA in nanopore as well as a concentrated electrolyte to amplify the volume-exclusion effect.

- a) 73.34% (by mass) or 88.56% (by volume) for 20 amino acids by AeL (*Nat. Biotechnol.* **2020**, 38, 176);
- b) 85.95% (by mass) for 7 peptides by FraC (*Nat. Commun.* **2021**, 12, 5795);
- c) 89.12% (by mass) for 4 proteins by YaxAB (*ACS Nano* **2023**, 17, 13685);
- d) 91.71% (by mass) for 7 proteins by ClyA (*ACS Nano* **2020**, 14, 2296);
- e) 93.43% (by mass) for 8 peptides by CytK (*ACS Nano* **2022**, 16, 7258).

Last but not least, we would like to express our sincere gratitude to you for such careful review and constructive comments. We wish that our response, supplementary work, and justification could fully address your concerns. As the modified title of our manuscript conveys, i.e., “*Machine learning assisted single-molecule sensing towards standard-free quantification of per- and polyfluoroalkyl carboxylic acids*”, the aim of this work is to provide proof-of-concept demonstration for our method, which in other reviewer’s word, is innovative and worth exploring. More examples of standard-free single-molecule identification and quantification of PFCAs isomers have been explored in our lab, and we are happy to include those results in the revision, if you and other reviewers request.

Reviewer #2 (Remarks to the Author):

Review Comments:

Key results

- The author have a sensor that they claim can “quantify a mixture of PFCAs with references.” They report the ability to predict molecular volumes from a linear relationship between current blockade and PFCa molecular volume.

Author Response:

Thank you for the time and efforts to review our recent submission, and your critical comments and suggestions to help improve the quality of our manuscript. It is with regret that, due to the length limit, some of the key findings and evidence of our work may not have been fully illustrated, which cast doubt on the novelty and practicability of this new approach. We would like to emphasize that *the proposed single-molecule sensing of PFCAs was a digital quantification method* (*Nature* **2024**, 628, 771; *Nat. Nanotechnol.* **2024**, 19, 800), with three consecutive breakthroughs:

- Standard-free prediction* of the blockade for unknown PFCAs using the linear volume-current relationship – this has been fully recognized by you and other reviewers;
- Standard-free identification* of a broad range of PFCAs via the high-resolution multi-feature classification. Notably, *the construction of feature library does NOT rely on PFCa standard either* – this is overlooked by most reviewers, and has been discussed in greater detail now. In short, since all the features are extracted from an individual single-molecule signal, the standard-free prediction of blockade is able to derive other related features, regardless of using pure or mixed PFCAs, as long as complete discrimination is realized. Our recent study confirmed that PFCa isomers could be accurately identified with this approach (<https://link.cnki.net/urlid/11.2097.X.20240927.1122.004>);
- Standard-free quantification* of multiple PFCAs in complex matrices using the probe-directed capture – *multiplexing in various environmental and biological interferences is actually a technically solved issue for single-molecule sensing*, but as most reviewers raise their concern on it, we have added 60 quantitative measurements with a total of 9 different interferences, including simultaneous quantifications of multiple species, to validate its feasibility. Moreover, we have provided a proof-of-concept demonstration that the capture rate of short-chain PFCAs is decided by the electrophoretic force on polycationic probes, enabling the universal single-molecule counting for absolute quantification.

Putting all these together, it is possible to *avoid the use of extraction standards for recovery calculation in complex matrices, injection standards for chromatography and mass spectrometry in identification*, and *native standards for building calibration curve in quantification*, which are all mandatory in conventional target analysis.

Scheme 1. Comparison of standard-free single-molecule sensing of PFCAs against LC-MS/MS.

Validity

- the rationale for the work is based on perceived limitations and technical challenges associated with current approaches for estimating concentrations of PFAS in humans and the environment.

Thank you for the correct interpretation of the rationale of our work!

Significance

- Quite a bit of the cited literature focuses on determinations of PFAS in environmental media, but the submitted manuscript does not attempt to demonstrate the technology in media of any complexity (e.g., electrolyte solution) and thus does not attempt to demonstrate the effectiveness of the technique in a system of any complexity. For example, natural waters contain organic matter, other trace organic and inorganic species. The suggestion in the manuscript is that this approach could replace current technology. The current manuscript does not bridge this gap. Thus, the significance of the technology is unclear.

Thank you for this critical comment. The initial emphasis of our manuscript is to highlight the development process towards standard-free quantification of PFCAs, but fails to fully illustrate the anti-interference performance in complex matrices. To address your concerns, a detailed workflow has been provided in Fig. 3a to better demonstrate the application of standard-free single-molecule quantification of PFCAs in realistic scenarios. Briefly, polycationic probes are first incubated with analytes to tether the PFCAs targets onto peptide leaders, which is a common pre-processing step in nanopore technology (<https://nanoporetech.com/products/prepare>), and has been verified in our own work as well (Fig. S40). Untethered inferences have a negligible effect on PFCAs analysis, no matter whether it is membrane stability, or identification and quantification performance (Fig. 3c, S36-37), while all those tethered PFCAs are determined simultaneously through multi-feature classification (Fig. 3b, S32-35). More specifically, we have performed 60 quantitative measurements with a broad range of common environmental or biological interferences, such as serum, tap water, urea, indole, ciprofloxacin, caprylic acid, TCEP, Pb^{2+} , and Cr^{2+} . These interferences caused insignificant change on the baseline current in most cases, and thus the major signal feature of PFCAs (e.g., blockade or dwell time) remained constant even when the concentrations of those interferences were 20 times greater (Fig. S36-37). Hence, the interference-free quantification was readily achievable (Fig. 3c).

Fig. 3 (a) The generalized workflow for probe-directed single-molecule sensing of PFCA with the assistance of machine learning. (b) Quantification of C2 in the presence of C4, C5 or C6 at 20°C and -50 mV. The concentration of C2 varied from 50 nM to 100 μM, while those of C4, C5 and C6 were kept at 6.7, 12.5 and 12.5 μM, respectively. (c) Interference-free quantification of C2 in the presence of serum or tap water at 20°C and -50 mV. The concentration of C2 varied from 500 nM to 100 μM.

Fig. S36 (a) Raw current traces without and with interference - urea, Tris(2-carboxyethyl)phosphine (TCEP), Pb^{2+} , indole, caprylic acid. (b-d) $\Delta I/I_0$, τ_{on} and τ_{off} of C0 at different concentrations of environmental or biological interferences. The concentration of C0 is 4 μM in urea and TCEP, and 2 μM in Pb^{2+} , indole and caprylic acid.

Fig. S37 (a) Raw current traces without and with interference - serum, TCEP, Cr^{2+} , Ciprofloxacin. (b-d) $\Delta I/I_0$, τ_{on} and τ_{off} of C2 at different concentrations of environmental or biological interferences.

The concentration of C2 is 2 μ M.

- Per- and polyfluoroalkyl substances (PFAS) are a broad diverse set of molecules. The conventional definition of PFCAs is perfluoroalkyl carboxylates. The authors define PFACs as per- and polyfluoroalkyl carboxylic acid. Thus, the authors do not appear to be using conventional nomenclature and for this reason their training set is unclear and they are not listed in any of the accompanying documents that I could find.

Thank you for the suggestion. We agree that the term “PFCAs” mostly refer to per-fluoroalkyl carboxylic acids or carboxylates, but it is also used as the abbreviation for per- and polyfluoroalkyl carboxylic acids in environmental research (*J. Hazard. Mater.* **2025**, 484, 136640; *J. Hazard. Mater.* **2024**, 464, 132937; *J. Hazard. Mater.* **2022**, 435, 128969). In this study, we first adopted the linear perfluoroalkyl carboxylic acids to establish the volume-blockade relationship, and then applied it to predict the current response of other polyfluoroalkyl carboxylic acids, and this was the reason why we chose the abbreviation of “PFCAs” for per- and polyfluoroalkyl carboxylic acids. To avoid the unnecessary confusion, we have summarized the chemical information of all the PFCAs used in this study in *Table S1*, including the labels of chemicals for training set.

Table S1. Summarized information of all PFCA analytes involved in this manuscript

Substance	Molecular Formula	CAS number	Abbreviation /Label
Trifluoroacetic acid	C ₂ HF ₃ O ₂	76-05-1	C2
Pentafluoropropionic acid	C ₃ HF ₅ O ₂	422-64-0	C3
Heptafluorobutyric acid	C ₄ HF ₇ O ₂	375-22-4	C4
Perfluorovaleric acid	C ₅ HF ₉ O ₂	2706-90-3	C5
Perfluorohexanoic acid	C ₆ HF ₁₁ O ₂	307-24-4	C6
Perfluoroheptanoic acid	C ₇ HF ₁₃ O ₂	375-85-9	C7
Perfluorooctanoic acid	C ₈ HF ₁₅ O ₂	335-67-1	C8
Perfluorononanoic acid	C ₉ HF ₁₇ O ₂	375-95-1	C9
3H-Tetrafluoropropionic acid	C ₃ H ₂ F ₄ O ₂	756-09-2	3H
3Cl-Tetrafluoropropionic acid	C ₃ HClF ₄ O ₂	661-82-5	3Cl
5H-Octafluoropentanoic acid	C ₅ H ₂ F ₈ O ₂	376-72-7	5H
3:3-Fluorotelomer carboxylic acid	C ₆ H ₅ F ₇ O ₂	356-02-5	FTA
7H-Dodecafluoroheptanoic acid	C ₇ H ₂ F ₁₂ O ₂	1546-95-8	7H
3,3,3-Trifluoro-2-hydroxypropanoic acid	C ₃ H ₃ F ₃ O ₃	684-07-1	2OH-D
3,3,3-Trifluoro-2-methylpropanoic acid	C ₄ H ₃ F ₃ O ₂	381-97-5	SYN28
3-(Trifluoromethyl)crotonic acid	C ₅ H ₃ F ₃ O ₂	69056-67-3	33
2,3,4,5-Tetrafluorobenzoic acid	C ₇ H ₂ F ₄ O ₂	1201-31-6	2345
2,6-Bis(trifluoromethyl)benzoic acid	C ₉ H ₄ F ₆ O ₂	24821-22-5	26

Data and methodology, Analytical approach

- Since I was unimpressed by the potential significance of the paper, I did not access the methodology.

The significance of standard-free single-molecule sensing of PFCAs has been strengthened in the revised manuscript. Hope it can get more of your attention this time.

Suggested improvements

- An important step would be to attempt to demonstrate the technology in something other than pure electrolyte solutions and the chemical training set for the measurements should be revealed

Thank you for the kind suggestions. We have examined the influence of common interferences in complex environmental matrices (*Fig. 3c, S36-37*), as mentioned in the above response. Besides, the simultaneous determination of two or more PFCAs has also been evaluated to further illustrate the broad practicability of our approach (*Workflow S3*), including typical examples of quantifications of C2 in the presence of C4/C5/C6 (*Fig. 3b, S32-34*), as well as detection of 3H or

3Cl with C5 (Fig. S35) and C5 with C2 (Workflow S3f). The list of chemical information for all the PFCAs used in this study is summarized in Table S1.

Fig. 3b Quantification of C2 in the presence of C4, C5 or C6 at 20°C and -50 mV. The concentration of C2 varied from 50 nM to 100 μ M, while those of C4, C5 and C6 were kept at 6.7, 12.5 and 12.5 μ M, respectively.

Fig. S35 Simultaneous quantification instances of (a) 3H and (b) 3Cl under the interference of C5-R6.

Workflow S3. Simultaneous quantification of mixed analytes.

References

- The references are fine but illustrate that the interest in these compounds centers on our ability to measure them in complex abiotic and biological systems.

Thank you for the positive comments on our references. The ability of single-molecule sensing of PFCAs in complex abiotic and biological systems has been extensively analyzed in the revision, as discussed in the previous response.

Reviewer #3 (Remarks to the Author):

Review Comments:

The manuscript with ID NCOMMS-24-39141-T by Zuo and colleagues reports an interesting study on the development of a machine learning-assisted method to identify and quantify per- and polyfluoroalkyl carboxylic acids (PFCA—a subfamily of PFAS) using a single-pore electrochemical system. The proposed method can quantify and identify PFCAs without the need of reference standards based on the unique electrochemical fingerprint of these molecules when they flow across the nanopore. The study is somehow comprehensive in the generation of a broad database of PFCA fingerprints by assisted machine learning approach. However, I have serious reservations about the actual utility of the proposed scheme for real-life scenarios. My main points of criticism are summarised as follows:

Author Response:

Thank you very much for your interest, the correct interpretation of our rationale, and positive comments on the potential significance for standard-free single-molecule quantification of PFCAs! The practicability of this new approach has been extensively examined in the revision. Hope it can fully address your concern now.

1 – The study uses polycationic peptide probes to control the location of tethered PFCA when these flow through the nanopore and induce the current blockade. This might be suitable for a laboratory study, where target PFCA is in its pure form. However, real-life PFCAs combine with other molecules in the environment. This alteration would introduce considerable errors in the identification.

Thank you for the critical comment. To better demonstrate the application of single-molecule sensing in realistic scenarios, a detailed workflow has been provided in *Fig. 3a*. Briefly, polycationic probes are first incubated with analytes to tether the PFCA targets onto peptide leaders, which is a common pre-processing step in nanopore technology (<https://nanoporetech.com/products/prepare>), and has been verified in our work as well (*Fig. S40*). Untethered inference has negligible effect on PFCA analysis, no matter whether it is membrane stability, identification or quantification accuracy (*Fig. 3c, S36-37*); and all those tethered PFCAs could be determined simultaneously through multi-feature classification (*Fig. 3b, S32-35*). The simultaneous determination of two or more PFCAs has also been evaluated to illustrate the wide practicability of our method (*Workflow S3*), including typical examples of quantifications of C2 in the presence of C4/C5/C6 (*Fig. 3b, S32-34*), and detection of 3H or 3Cl with C5 (*Fig. S35*) and C5 with C2 as well (*Workflow S3f*). Besides, the prediction accuracy of the established linear volume-blockade relationship is further proven by five distinct Poly-FCAs containing methyl/hydroxyl side chains, double bonds or aromatic groups (*Fig. S6*).

Fig. 3 (a) The generalized workflow for probe-directed single-molecule sensing of PFCA with the assistance of machine learning. (b) Quantification of C2 in the presence of C4, C5 or C6 at 20 °C and -50 mV. The concentration of C2 varied from 50 nM to 100 μM, while those of C4, C5 and C6 were kept at 6.7, 12.5 and 12.5 μM, respectively. (c) Interference-free quantification of C2 in the presence of serum or tap water at 20 °C and -50 mV. The concentration of C2 varied from 500 nM to 100 μM.

Fig. S35 Simultaneous quantification instances of (a) 3H and (b) 3Cl under the interference of C5-R₆.

Workflow S3. Simultaneous quantification of mixed analytes.

Fig. S6 Experimentally measured current blockade and molecular volume of poly-fluoroalkyl carboxylic acids with other structures, along with the established volume-current relationship (grey line).

2 – Another important aspect of is the use of highly concentrated electrolytes to control the surface charge. One of the biggest issues in the identification of PFCAs (and more broadly PFAS) is the interferences introduced by different water matrices with varying composition and ionic strength. While I appreciate the technical merit of the study (I think that the approach is innovative and worth exploring), I do question its practical applicability. The use of standards is still a necessity to elucidate the effect of electrolyte and composition of PFAS in real-life water samples.

Thank you for the valuable comments. The concentrated electrolyte is a common approach in nanopore technology (e.g., commercial DNA/RNA sequencing by Oxford Nanopore Technologies) to enhance the resolution for identification. To validate the reliability of single-molecule sensing of PFCAs in complex environmental matrices, we have performed 60 quantitative measurements with a broad range of common environmental or biological interferences, such as serum, tap water, urea, indole, ciprofloxacin, caprylic acid, TCEP, Pb^{2+} , and Cr^{2+} . In most cases, these interferences caused insignificant change on baseline current, and thus the major signal features of C2/C0 (e.g., blockade or dwell time) remained constant even when the concentrations of those interferences were 20 times greater (Fig. S36-37). Hence, the interference-free quantification was readily achievable (Fig. 3c).

We would like to emphasize that the proposed single-molecule sensing of PFCAs was a digital quantification method (*Nature* **2024**, 628, 771; *Nat. Nanotechnol.* **2024**, 19, 800; *Nat. Commun.* **2023**, 14, 653), with three consecutive breakthroughs: a) standard-free prediction of the blockade for unknown PFCAs using a strict linear volume-current relationship; b) standard-free identification of a wide range of PFCAs via the high-resolution multi-feature classification; c) standard-free quantification of multiple PFCAs in complex matrices using the probe-directed capture. Putting all these together, it is thus possible to avoid the use of extraction standards for recovery calculation in complex matrices, injection standards for LC/MS-MS or GC in identification, and native standards for building calibration curve in quantification, which are all mandatory in conventional target analysis. The feasibility of standard-free prediction, identification and quantification have been thoroughly examined in the revised manuscript, which hopefully can convince you this time.

Fig. 3 (c) Interference-free quantification of C2 in the presence of serum or tap water at 20°C and -50 mV. The concentration of C2 varied from 500 nM to 100 μM.

Fig. S36 (a) Raw current traces without and with interference - urea, Tris(2-carboxyethyl)phosphine (TCEP), Pb^{2+} , indole, caprylic acid. (b-d) $\Delta I/I_0$, τ_{on} and τ_{off} of C0 at different concentrations of environmental or biological interferences. The concentration of C0 is 4 μM in urea and TCEP, and 2 μM in Pb^{2+} , indole and caprylic acid.

Fig. S37 (a) Raw current traces without and with interference - serum, TCEP, Cr²⁺, Ciprofloxacin. (b-d) I/I_0 , τ_{on} and τ_{off} of C2 at different concentrations of environmental or biological interferences. The concentration of C2 is 2 μ M.

Scheme 1. Comparison of standard-free single-molecule sensing of PFCAs against LC-MS/MS.

Remarks on code availability:

The code is usable by the community

Thank you for the time and efforts to test our code.

Reviewer #4 (Remarks to the Author):

Review Comments:

Environmental contamination by perfluoroalkyl and polyfluoroalkyl substances (PFAS) is a notable global concern. Current methods for detecting PFAS typically require standard samples, which are limited in availability. Therefore, methods that do not rely on standard samples are highly desirable. The authors proposed using wild-type aerolysin nanopores in a 4-M KCl solution for detecting perfluorocarboxylic acids (PFCAs) without standard samples. They reported a strong linear relationship between molecular volume and current and achieved almost 100% accuracy in identifying PFCAs using the bag tree classification model. They also demonstrated that PFCAs can be quantified even in the presence of interference.

Author Response:

Thank you very much for the correct interpretation of our rationale, and positive comments on the potential significance towards standard-free single-molecule sensing of PFCAs! Due to the limit of length, some of the key findings and evidence of our work may not have been fully illustrated in the initial submission, which cast doubt on the novelty and practicability of this new approach. It is worth noting that *the proposed single-molecule sensing of PFCAs is a digital quantification method* (*Nature* **2024**, 628, 771; *Nat. Nanotechnol.* **2024**, 19, 800), with three consecutive breakthroughs:

- Standard-free prediction* of the blockade for unknown PFCAs using the linear volume-current relationship – this has been fully recognized by you and other reviewers;
- Standard-free identification* of a broad range of PFCAs via the high-resolution multi-feature classification. Notably, *the construction of feature library does NOT rely on PFCA standard either* – this is overlooked by most reviewers, and has been discussed in greater detail now. In short, since all the features are extracted from an individual single-molecule signal, the standard-free prediction of blockade is able to derive other related features, regardless of using pure or mixed PFCAs, as long as complete discrimination is realized. Our recent study confirmed that PFCA isomers could be accurately identified with this approach (<https://link.cnki.net/urlid/11.2097.X.20240927.1122.004>);
- Standard-free quantification* of multiple PFCAs in complex matrices using the probe-directed capture – *multiplexing in various environmental and biological interferences is actually a technically solved issue for single-molecule sensing*, but as most reviewers raise their concern on it, we have added 60 quantitative measurements with a total of 9 different interferences, including simultaneous quantifications of multiple species, to validate its feasibility. Moreover, we have provided a proof-of-concept demonstration that the capture rate of short-chain PFCAs is decided by the electrophoretic force on polycationic probes, enabling the universal single-molecule counting for absolute quantification.

Scheme 1. Comparison of standard-free single-molecule sensing of PFCAs against LC-MS/MS.

Putting all these together, it is possible to *avoid the use of extraction standards for recovery calculation in complex matrices, injection standards for chromatography and mass spectrometry in identification, and native standards for building calibration curve in quantification*, which are all mandatory in conventional target analysis.

1. Although the introduction aims to quantify PFAS without relying on standard samples, the paper does not fully address this issue. The authors mention that there are over 14,000 PFAS chemical species but only about 120 standard samples. Their study focuses on nanopore measurements of PFCAs (C2–C9) and some of their substituents. Since most PFAS molecules lack carboxyl groups, even if the method accurately predicts PFCAs with carboxyl groups, its applicability to PFAS without carboxyl groups is uncertain. This paper does not demonstrate a method that can be applied to molecules without carboxyl groups, leaving the initial problem unsolved.

Thank you for this critical question. A detailed workflow has been provided in *Fig. 3a* to better illustrate the wide applicability of standard-free single-molecule quantification of PFCAs in realistic scenarios. Briefly, polycationic probes are first incubated with analyte to tether the PFCA targets on peptide leaders, and thus for PFAS that does not contain a carboxyl terminal group, it is also facile to design other types of probes with different recognition domains. This is a common pre-processing step in nanopore technology (<https://nanoporetech.com/products/prepare>), and has been verified in our work as well (*Fig. S40*). Untethered inference has negligible effect on PFCA analysis, no matter whether it is membrane stability, identification or quantification accuracy (*Fig. 3c, S36-37*); while all the tethered PFCAs are determined simultaneously through multi-feature classification (*Fig. 3b, S32-35*).

Fig. 3 (a) The generalized workflow for probe-directed single-molecule sensing of PFCA with the assistance of machine learning. (b) Quantification of C2 in the presence of C4, C5 or C6 at 20 °C and -50 mV. The concentration of C2 varied from 50 nM to 100 μM , while those of C4, C5 and C6 were kept at 6.7, 12.5 and 12.5 μM , respectively. (c) Interference-free quantification of C2 in the presence of serum or tap water at 20 °C and -50 mV. The concentration of C2 varied from 500 nM to 100 μM .

Fig. S40 Typical blockade and standard deviation signals for (a) the R₆ probes (C0) and (b) the incubation of R₆ probes with different concentrations of activated C2/C4/C6. The overall duration for activation and incubation is roughly 1 hour.

2. The authors achieved high classification accuracy by constructing a machine learning model with multiple features. However, supervised learning typically requires standard samples. To obtain measurements without relying on standard samples, it may be more appropriate to develop a regression model that predicts the properties of molecules not included in the training data. For instance, a regression model could estimate molecular volume or carbon number using the same feature set. However, even with such analysis, eliminating the need for standard PFAS samples may still be not feasible.

Thank you for this valuable suggestion! However, the construction of feature library does NOT rely on PFCA standards, as mentioned in the response above. In fact, even when pure PFCA samples (not necessarily standards) are commercially available, they need to be measured with some other known PFCA, e.g., C3/C5/C6 in this work, to calibrate their blockade and reduce experimental errors (*Workflow S1*). With regard to unknown PFCAs (*Workflow S2*): a) the frequency-modulated multiple dimensional features of all current events were extracted, including the signals for calibration PFCA; b) unsupervised clustering was applied to separate and aggregate the current events of unknown PFCAs into certain groups; c) the calibrated blockade of each group was compared to the volume-current relationship (*Fig. S41*) for standard-free identification; and d) other multi-dimensional features of the identified PFCA were labelled and added into the signal database. Hence, the ultra-high resolution of the multiple-feature classification and clustering is crucial, and we have confirmed recently that PFCA isomers could be accurately identified with this approach.

Workflow S1. Conventional data acquisition, feature extraction and model training with known

PFCA samples.

Workflow S2. Standard-free identification and feature extraction of unknown PFCA.

3. The study uses the linearity between excluded volume and current to support a quantification method without standard samples. Although linear chains of PFCAs have been successfully measured and predicted, branched-chain PFCAs also exist. For example, can PFCAs having a trifluoromethyl group ($-\text{CF}_3$) be classified using the same approach?

Thank you very much for this insight suggestion. Five more other types of PFCAs have been measured in the revision to confirm the wide applicability of the volume-current relation established in this work (Fig. S6), i.e., one with a methyl side-chain (3,3,3-trifluoro-2-methylpropanoic acid), another one with a methyl side-chain and an unsaturated bond (3-(trifluoromethyl)crotonic acid), one with a hydroxyl side-chain (3,3,3-trifluoro-2-hydroxypropanoic acid), and two with aromatic groups (2,3,4,5-tetrafluorobenzoic acid and 2,6-bis(trifluoromethyl)benzoic acid), confirming the.

Figure S6. Experimentally measured current blockade and molecular volume of poly-fluoroalkyl carboxylic acids with other structures, along with the established volume-current relationship (grey line).

4. The model was validated only for PFCA molecules up to C9. Can the model be applied to longer molecules? For instance, C14 is regulated by the European Union; can the model be extrapolated to accommodate C14?

Thank you for this critical question. Theoretically speaking, the linear range of volume-current relationship was decided by the slope of this straight line (1.68% per $-\text{CF}_2-$ or $0.023\% \cdot \text{\AA}^{-3}$), and the effective transduction volume between the A224 and S236 residues of WT AeL (4.82 nm^3), *Fig. 1d*. Hence, it is possible to extend our detection range to C14, which possesses a blockade of 71.48%. However, it is also noted when the length of PFCA further increases from C5 to C8, their respective interval time is prolonged (*Fig. S38*), indicating that the capture rate of C14 may be extremely slow. This is why we put more emphasis on ultra- and short-chain PFCA in this study. We are working on nanopore engineering (e.g., through site-specific mutation of amino acids at the entrance of AeL) to further enhance the capture rate of PFCA. Wish we can demonstrate those results in the following work soon.

Fig. 1 (d) The established linear relationship between the hydrodynamic volume of C0/C2-C9 PFCA- R_6 (blue squares and straight line) and the magnitude of their current blockade. The volume of PFCA was calculated via MD simulation using GROMACS (Figures S2-S3). The experimentally measured current blockade of 3H, 5H, 7H, 3Cl, and FTA was shown as brown circles, which was almost identical with the prediction from volume-current correlation (insert in Figure 1d). The error bars on blue squares and brown circles were the sum of three times standard deviations (NOT one) obtained from the histogram of their current blockade.

Fig. S38 Comparison of the interval time for C0 and C2 to C8 at -50 mV.

5. The authors attribute the observed linearity to polycationic peptide probes and concentrated electrolytes. However, could this linearity be a molecular characteristic? Is similar linearity observed in non-fluorinated molecules?

Thank you for this critical question. The possibility to establish a linear relationship between the volumes of analyte and the magnitude of current blockade has been explored in recent nanopore study, but our work has realized for the first time a *strict* linear correlation (i.e., $R^2 = 99.98\%$ for 13 PFCAs), which is the prerequisite for standard-free prediction and identification. In contrast, other latest progress of single-molecule analysis only reached R^2 of 73.34% to 93.43% based on the mass of amino acids, peptides, or proteins (see below for details), despite the volume-current relationship is much sounder theoretically (*Nat. Biotechnol.* **2020**, 38, 176). As you mentioned, key innovations are the use of a polycationic probe to control the position of tethered PFCA in nanopore as well as a concentrated electrolyte to amplify the volume-exclusion effect.

- a) 73.34% (by mass) or 88.56% (by volume) for 20 amino acids by AeL (*Nat. Biotechnol.* **2020**, 38, 176);
- b) 85.95% (by mass) for 7 peptides by FraC (*Nat. Commun.* **2021**, 12, 5795);
- c) 89.12% (by mass) for 4 proteins by YaxAB (*ACS Nano* **2023**, 17, 13685);
- d) 91.71% (by mass) for 7 proteins by ClyA (*ACS Nano* **2020**, 14, 2296);
- e) 93.43% (by mass) for 8 peptides by CytK (*ACS Nano* **2022**, 16, 7258).

6. The environment may contain fatty acids that could interfere with the measurements. Is the method still effective if fatty acids with similar volumes are present?

Thank you for this critical comment. To validate the reliability of single-molecule sensing of PFCAs in complex environmental matrices, we have performed 60 quantitative measurements with a variety of common environmental or biological interferences (including fatty acids, e.g., caprylic acid). In most cases, these interferences caused insignificant change on the baseline current, and thus the major signal features of C2/C0 (e.g., blockade and dwell time) remained constant even when the concentrations of those interferences were 20 times greater (*Fig. S36-37*). As a result, the interference-free quantification was readily achievable (*Fig. 3c*).

Fig. 3 (c) Interference-free quantification of C2 in the presence of serum or tap water at 20°C and -50 mV. The concentration of C2 varied from 500 nM to 100 μM.

Fig. S36 (a) Raw current traces without and with interference - urea, Tris(2-carboxyethyl)phosphine (TCEP), Pb^{2+} , indole, caprylic acid. (b-d) $\Delta I/I_0$, τ_{on} and τ_{off} of C0 at different concentrations of environmental or biological interferences. The concentration of C0 is 4 μM in urea and TCEP, and 2 μM in Pb^{2+} , indole and caprylic acid.

Fig S37. (a) Raw current traces without and with interference - serum, TCEP, Cr²⁺, Ciprofloxacin. (b-d) $\Delta I/I_0$, τ_{on} and τ_{off} of C2 at different concentrations of environmental or biological interferences. The concentration of C2 is 2 μ M.

7. A reproducible description of the analysis method or the source code should be made publicly available, including details of the mother wavelet used. Because nanopores are laboratory-fabricated rather than commercially produced, publicly available measurement data can enhance the reliability of the obtained results, particularly in research that integrates nanopores with artificial intelligence.

Thank you very much for this kind suggestion. We have supplemented a series of workflows explaining our analysis methods in detail at the beginning of Supplementary Information. All the custom MATLAB and Python scripts are available at https://github.com/DEWMEME/Frequency-modulated-multi-dimensional-feature-extraction-for-PFCAs-signal_V2/tree/master. Moreover, the statistical results of 43 features extracted from 13 PFCAs are given in Fig. S8-21, which have been used as the training set in the evaluation of model performance for classification. All the source data have been provided with the paper, and other data are available upon request.

Response Letter

Reviewer #1 (Remarks to the Author):

Review Comments:

I am truly grateful for the author's efforts in responding to my comments. Many of my comments have also received responses. However, the evidence that supporting methods can standard-freely quantify multiple PFAS under real environmental conditions is still insufficient.

Author Response:

Thank you very much for the recognition of our efforts in the last round of revision, and many new insightful comments and suggestions to help improve the quality of our manuscript. In order to fully address your concerns, we have designed and implemented a large number of experiments. We are now very confident that the machine-learning assisted, probe-directed nanopore single-molecule sensor is capable of a) standard-free quantification of b) multiple PFCA simultaneously, c) under real environmental conditions. As re-stated explicitly in the latest version of revised manuscript, a unified workflow for single-molecule sensing of PFCA included four steps (Fig. 3a):

Figure 3. a The unified workflow for probe-directed single-molecule sensing of PFCA with the assistance of machine learning

- i) *PFCA were collected, activated, and tethered to polycationic peptide probes.* In this step, only PFCA and non-PFCA carboxylic acids could be connected to probes (Fig. S53), while other kinds of interference remained untethered in solution;
- ii) *Mixture of tethered PFCA were transferred to nanopore measurements,* as well as other tethered non-PFCA carboxylic acids and untethered interference. In this step, the untethered interference either had a negligible effect on measurements, or caused tiny signals that would not impact PFCA identification (Fig. S41-S42), while the tethered non-PFCA carboxylic acids could also be easily distinguished from PFCA (Fig. S40);
- iii) *Accurate, standard-free identification of each individual single-molecule signal was realized for a broad range of PFCA simultaneously by multi-feature classification.* It was shown that high-resolution identification of diverse, structurally similar PFCA was facile based on our customized signal feature library (Fig. 2 and S50-S51), as well as discrimination of non-PFCA carboxylic acids (even with identical blockade, Fig. S32), and the construction of feature library did not rely on authentic standards (as explained in Workflow S1-S2);
- iv) *Standard-free digital quantification of various PFCA, by counting their respective single-molecule signals, was accomplished through a “one-calibration curve-fil-all”*

strategy. As a proof-of-concept, the combination of -R₆ probe and R282A mutant pore enabled a universal capture rate up to C6 (Fig. 4e), covering almost 50% of the known PFCAs, and the slope of linear volume-current relation remained unchanged (0.023% Å⁻³), compared to wild type aerolysin (Fig. 4g and S45).

According to the above process, brief answers to those three major concerns were given below, and a more detailed point-by-point response to your specific questions was provided afterwards.

a) Standard-free quantification

Based on the digital measurement principle of nanopore single-molecule sensor, standard-free identification and quantification of PFCAs were investigated sequentially in this study. First of all, standard-free identification was done in two steps, i.e., standard-free prediction of signal features for unknown PFCAs based on linear volume-blockade relationship, followed by high-resolution identification through multiple-feature classification with our in-house feature library. After that, standard-free quantification was fulfilled by a “one-calibration curve-fit-all” strategy, or probe-directed universal captures rates for various PFCAs.

In this revised manuscript, the linear volume-current relationship was successfully extended to C14 using a R282A mutant aerolysin nanopore at -70 mV and 30°C (Fig. 4g), which meant that over 97% of PFCAs' blockade could be correctly predicted without standards. It was also found that both R282A and WT pore shared an identical slope of 0.023% Å⁻³, no matter at a mild or elevated voltage and temperature (Fig. 1d and S45). Meanwhile, multi-feature classification has proven effective to enhance the accuracy of PFCAs identification, even for targets or interferences of indistinguishable blockade (Fig. 2 and S32). Putting together the standard-free current prediction and high-resolution multiple-feature discrimination, it was thus possible to construct the signal feature library for PFCA identification without using authentic standards. In a typical PFCAs measurement by nanopore (e.g., the mixture of analytes contained various tethered PFCAs, tethered non-PFCA carboxylic acids, and untethered interferences, Fig. 3a): the untethered interferences had little effect on current responses as mentioned earlier; the non-PFCA carboxylic acids exhibited shorter τ_{on} and greater I_{σ} , compared to PFCAs of similar blockade (Fig. S32), and could be easily excluded; the PFCAs of known single-molecule signals were identified via the pre-established feature library (Fig. 2c and S46); and finally, for PFCAs with known structures but not included in the feature library, they were double confirmed by volume-blockade conversion and pre-knowledge on their approximate abundance in the mixture. As illustrated in this example, high-purity PFCA standards were dispensable for identification since all the individual single-molecule signals of PFCAs (with known structures or blockades) were well resolved from each other or interferences. Signals of new PFCAs would then be added to the feature library. So far, we have experimentally obtained full signal features of over 50 PFCAs (including 8 classes of 30 aliphatic or aromatic isomers⁴⁹⁻⁵², Fig. S50-S51), which accounted for 8% of the total PFCAs. Based on this feature library, our preliminary results confirmed that those diverse isomers with tiny structural variations could be easily differentiated in 100% accuracy (*Env. Eng.*, **2025**; *Env. Chem.*, **2025**), and we are now working on the standard-free prediction of full feature library for all 622 PFCAs.

In addition, the “one-calibration curve-fit-all” strategy for standard-free quantification was also extended to median-chain C6 in the revised manuscript by R282A mutant aerolysin nanopore (Fig. 4e), covering 50% of the total PFCAs. Steered molecular dynamics (SMD) analysis showed that the entry barrier of PFCA-R₆ complex into R282A nanopore was significantly lower than WT aerolysin

(Fig. 4c-4d). The universal capture rates or entry barriers of R₆ and C2/C3/C4/C5/C6-R₆ conjugates by R282A strongly supported the argument that the captures of PFCAs must be probe-controlled or analyte-independent when their entry barrier into AeL nanopore was trivial, compared to driving forces. Moreover, the exit barrier of R282A mutant remained unchanged, indicating that the linear volume-blockade relation (Fig. S45) and excellent identification performance (Fig. S46) for PFCAs-R₆ complex were uncompromised during the optimization of standard-free quantification. In order to further demonstrate the possibility of fulfilling standard-free quantification for long-chain PFCAs, we simulated the nanopore-analyte interaction distribution of R282S/E216S/R220S/E222S/A260F, a heavily engineered nanopore with a considerably reduced entry barrier for PFCAs-R₆ complex and an enhanced exit barrier as well (via cation- π interactions). It was found that, in this particular case, the universal capture rates would be able to cover C14 (Fig. S48), corresponding to 97% of the total PFCAs. The above evidence confirmed the feasibility of precisely tuning the entry and exit barriers of aerolysin nanopore to independently improve the dwell time and capture rate of PFCAs towards the complete standard-free quantification of these emerging contaminants.

Figure 4. Optimizing the double-barriers of probe-pore interaction towards directed evolution of standard-free single-molecule sensing of PFCAs. **a, b** Top-down views of the cis entry for WT and R282A aerolysin nanopores, highlighting the structures of arginine or alanine residue at the 282 position, and the respective electrostatic potential distributions estimated at 4 M KCl, 20°C and -50 mV. **c, d** SMD simulations for C0/C2/C4/C6-R₆ from cis entry to trans exit through WT and R282A aerolysin nanopores. **e** Voltage-dependent capture rates of C0/C2/C3/C4/C5/C6-R₆ by R282A under the same concentration. **f** Calibration curve of C2, measured with R282A at 30°C and -70 mV. The

concentration of C2 varied from 0.1 to 5 nM. **g** The linear volume-blockade relation measured with R282A at 30°C and -70 mV, covering long-chain PFCAs up to C14.

Figure 1. d The established linear relationship between the hydrodynamic volume of C0/C2-C9 PFCA-R₆ and the magnitude of their current blockade (blue squares and straight line). The volumes of PFCAs were calculated via MD simulation using GROMACS (Supplementary Fig. 2 and 3). The experimentally measured current blockade of 3H, 5H, 7H, 3Cl, and FTA was shown as brown circles, which was almost identical with the prediction from volume-current correlation (insert in Fig. 1d). The error bars on blue squares and brown circles were the sum of three times standard deviations (NOT one) obtained from the histogram of their current blockade.

Figure S45. The established linear relationship between the hydrodynamic volume of C0-C9 PFCA-R₆ and the magnitude of their current blockade (a) by WT AeL at 30°C and -80 mV, and (b) by R282A AeL at 20°C and -40 mV.

Figure S48. (a) Required force to pull C0/C2/C4/C6/C8-R₆ conjugate through R282A aerolysin nanopore based on steered molecular dynamics (SMD). (b) Required force to pull C6-R₆ conjugate through WT, R282A, or R282S/D216S/R220S/D222S/A260F (4S1F) aerolysin nanopore based on SMD. (c) Required force to pull C6/C8/C10/C12/C14-R₆ conjugate through 4S1F aerolysin nanopore based on SMD.

b) Multiple PFCAs simultaneously

Given that nanopore single-molecule sensing is a digital quantification approach that is mainly

performed in dilute solutions, it is readily achievable to measure multiple PFCAs simultaneously as the overall capture rate is simply the sum of their individual values. For instance, if there are i types of PFCAs in the mixture, the total capture rate k is expressed as $k = \sum k_i$, and the number of signals for each analyte is in proportion to their individual capture rates ($N_p/N_q = k_p/k_q$), or is inversely proportional to their ‘true’ interval time ($N_p/N_q = t_{off}^q/t_{off}^p$). In this regard, the total measurement time t equals to $t = \sum N_i \times (t_{off} + t_{on}^i)$, where t_{off} is the observed interval time, and t_{on}^i is the analyte-specific dwell time that is independent of their concentrations. On the other hand, the ‘true’ analyte-specific interval time is inversely proportional to their concentrations, and is calculated as $t_{off}^i = t_{off} \times \sum N_i/N_i$. To elucidate this theory, we have measured calibration curves of 10 PFCAs in the presence of internal standards of fixed concentrations (Fig. S35-S39). These PFCAs included C2-C6 and 3H/3Cl/5H/7H/FTA, and the internal standards were mainly C3-C7. It was seen that the ‘true’ interval times of internal standards remained constant throughout the process, and the resulting calibration curves of all measured PFCAs were indistinguishable no matter whether in the presence or absence of (any kind of) internal standards. Hence, as long as the standard-free identification and quantification can be achieved, simultaneous determination of multiple PFCAs is straightforward.

Figure S35. (a-d) Representative current traces (left) and signal intensity histograms (right) of C2-

R₆ with different concentration (1~10 μM) under the interference of C4-R₆.

Figure S36. (a-e) Representative current traces (left) and signal intensity histograms (right) of C2-R₆ with different concentration (1~20 μM) under the interference of C5-R₆.

Figure S37. (a-e) Representative current traces (left) and signal intensity histograms (right) of C2-R₆ with different concentration (1~20 μM) under the interference of C6-R₆.

Figure S38. Simultaneous quantification instances of (a) 3H with C5-R₆ interference, (b) 3Cl with C5-R₆ interference, (c) 5H with C6-R₆ interference, (d) C5 with C3-R₆ interference, (e) C5 with C7-R₆ interference, (f) FTA with C5-R₆ interference, and (g) 7H with C5-R₆ interference.

Figure S39. Calibration curves of C3, C4, 5H, C5 and C6 measured in the presence of internal standards of fixed concentrations.

c) Under real environmental conditions

To illustrate the reliability of nanopore single-molecule sensing of PFCAs under a broad range of real environment/biological interferences, including the serum and tap water matrices, untethered urea, indole, ciprofloxacin, TCEP, Pb^{2+} or Cr^{2+} , and tethered fatty acids as well. As explained above or in the previous revision, the untethered interferences generated no signals in most cases, and thus had insignificant impact on the baseline current, main signal features (e.g., blockade or dwell time), and calibration curves of target PFCAs (e.g., C2-C4), even when the concentration of interferences was 20 times greater (Fig. 3f, S41-S43). The tethered fatty acids, though causing non-negligible or even comparable magnitude of current blockade with PFCAs, their other signal features (e.g., dwell time or standard deviation of current response) were easily distinguishable via clustering (Fig. S32) and would not affect the quantification of target PFCAs (Fig. S40). Therefore, common interferences affect little on the standard-free identification and quantification performance of PFCAs, and it is practicable to conduct nanopore single-molecule sensing under real environmental conditions.

Figure 3. f Calibration curves of C2 in the presence of serum or tap water. The concentration of C2

varied from 500 nM to 100 μ M.

Figure S43. Interference-free quantification of (a) C3 and (b) C4 in presence of tap water or serum at 20°C and -50 mV.

Figure S41. (a) Raw current traces without and with untethered interference - urea, Tris(2-carboxyethyl)phosphine (TCEP), Pb^{2+} , indole, caprylic acid. (b-d) $\Delta I/I_0$, τ_{on} and τ_{off} of C0 at different concentrations of environmental or biological interferences. The concentration of C0 was 4 μ M in urea and TCEP, and 2 μ M in Pb^{2+} , indole and caprylic acid.

Figure S42. (a) Raw current traces without and with untethered interference - serum, TCEP, Cr²⁺, Ciprofloxacin. (b-d) $\Delta I/I_0$, τ_{on} and τ_{off} of C2 at different concentrations of environmental or biological interferences. The concentration of C2 was 2 μM.

Figure S40. Quantification of C2-R₆ under tethered interference of valeric acid (FA5-R₆).

Figure S32. (a) One-, (b) two- and (c) three-dimensional distribution of cluster results of C3 and butyric acid (FA4), whose current blockades are extremely close. (d) Mean values of molecular volumes, $\Delta I/I_0$, I_σ , τ_{on} and H_{peak} of C3 and FA4. The concentration ratio of C3 and FA4 in mixed samples was set as 4:1, and the signal number ratio achieved by unsupervised clustering was 3.89:1.

1. I am concerned about the generality of the standard-free quantification. In standard free quantification, the author used ultra short chain and short chain PFCAs (C2/C3/C4) as demonstrations, and they also provided the voltage dependent capture pattern of C0, C2, C3 and C4. But how about the capture pattern along with the increasing of their concentration? The author also give reason for this, “as the electrostatic driving force on polycationic peptide was much greater than the entry barrier into AeL nanopore for ultrashort- or short-chain PFCA.” But how about the capture pattern of medium and long chain PFCAs? These results will impact whether one concentration-interval time response curve fit all different kinds of PFCA.

Thank you for your suggestion. We have included the concentration-dependent capture patterns of C2, C3 and C4 in the revision. It was clear that, under the same applied voltage of -50 mV, these three PFCAs exhibited highly consistent concentration-interval time response curves (Fig. S34), in line with their identical voltage-driven capture rates (Fig. 3b). In other words, if a group of PFCAs showed similar voltage-dependent capture patterns at the same concentrations, they would probably possess comparable calibration curves. This was because the analyte capture process was composed of three steps (i.e., diffusion, migration, and entry), in which the frequency of analyte diffusion and migration to nanopore opening was proportional to analyte concentration and applied voltage, while the entry ‘probability’ of analyte into nanopore increased exponentially with voltage (*J. Electrochem. Soc.*, **2021**, *168*, 126502). Therefore, the voltage-driven capture pattern could provide more insights into the capture mechanisms of various PFCAs. For example, the capture kinetics (the reciprocal of interval time) of C2 and C8 by WT AeL nanopore suggested that the former was diffusion-limited while the latter was barrier-limited, confirming that the capture of ultrashort- or short-chain PFCA-R₆ was mainly decided by the driving force on polycationic peptide probe (R₆) and the entry barrier into WT AeL nanopore was negligible. Based on the above considerations, we adopted the voltage-dependent capture patterns in the comparison of various PFCA/nanopore combinations.

Figure S34. Shared concentration-interval time response curve of C2, C3 and C4.

Figure 3. b Voltage-dependent capture rates for C2, C3 and C4, as well as the R₆ probe, under the same concentration.

To further extend the universal capture rate and standard-free quantification to median- or long-chain PFCAs, we attempted to lower down the entry barrier of aerolysin nanopore by replacing the first positive-charge arginine at the cis opening with a shorter-chain hydrophobic alanine. As shown in all-atom molecular dynamics simulation, the resulting R282A mutant could effectively modulate the electrostatic potential distribution around the entry of aerolysin nanopore (Fig. 4a-4b), switching the repulsion force against R₆ probe into attraction. The steered molecular dynamics (SMD) analysis confirmed that the forces required to pull PFCa-R₆ conjugates at a constant speed through nanopore, or the energy barriers to overcome, were consistently smaller at the entry of R282A than WT. More specifically, it was found that the entry barriers of WT AeL were comparable for the capture of C0, C2 and C4, but considerably higher for C6 (Fig. 4c), in line with the experimental observations. In contrast, the R282A mutant exhibited constantly low entry barriers for C6 and C0/C2/C4 (Fig. 4d). Inspired by this, we then experimentally examined the voltage-driven capture patterns of C5 and C6 by R282A, which was unsurprisingly akin to those of C2/C3/C4 and R₆ probe (Fig. 4e), expanding the coverage of standard-free quantification to almost 50% of PFCAs. The elevated capture rates by R282A mutant also improved the LOD of C2 by five times, down to 11.4 ng·L⁻¹ or 0.1 nM (Fig. 4f), which was among the best of all existing techniques. Most importantly, it was seen that the volume-blockade relationship of PFCAs remained linear for R282A mutant, which shared an identical slope of 0.023% · Å⁻³ with WT AeL, no matter at mild or elevated voltages and temperatures (Fig. 1d, 4g, S45). This phenomenon supported our hypothesis that standard-free quantification or identification could be independently tuned as they were separately controlled by the entry or exit barrier of probe-pore interactions. Besides, the linear volume-blockade relationship of R282A mutant was extended to C14, due to the smaller magnitude of blockade induced by greater open-pore current and driving force, which was able to cover more than 97% of PFCAs. In short, our experimental and simulation results proved that the R282A mutant was likely to realize standard-free identification for all PFCAs and quantification for half of them.

Figure 4. Optimizing the double-barriers of probe-pore interaction towards directed evolution of standard-free single-molecule sensing of PFCAs. **a,b** Top-down views of the cis entry for WT and R282A aerolysin nanopores, highlighting the structures of arginine or alanine residue at the 282 position, and the respective electrostatic potential distributions estimated at 4 M KCl, 20°C and -50 mV. **c,d** SMD simulations for C0/C2/C4/C6-R₆ from cis entry to trans exit through WT and R282A aerolysin nanopores. **e** Voltage-dependent capture rates of C0/C2/C3/C4/C5/C6-R₆ by R282A under the same concentration. **f** Calibration curve of C2, measured with R282A at 30°C and -70 mV. The concentration of C2 varied from 0.1 to 5 nM. **g** The linear volume-blockade relation measured with R282A at 30°C and -70 mV, covering long-chain PFCAs up to C14.

Figure 1. d The established linear relationship between the hydrodynamic volume of C0/C2-C9 PFCAs and the magnitude of their current blockade (blue squares and straight line). The volumes

of PFCAs were calculated via MD simulation using GROMACS (Supplementary Fig. 2 and 3). The experimentally measured current blockade of 3H, 5H, 7H, 3Cl, and FTA was shown as brown circles, which was almost identical with the prediction from volume-current correlation (insert in Fig. 1d). The error bars on blue squares and brown circles were the sum of three times standard deviations (NOT one) obtained from the histogram of their current blockade.

Figure S45. The established linear relationship between the hydrodynamic volume of C0-C9 PFCA-R₆ and the magnitude of their current blockade (a) by WT AeL at 30°C and -80 mV, and (b) by R282A AeL at 20°C and -40 mV.

To explore the question whether the nanopore based single-molecule sensors would be capable of standard-free quantification of all PFCAs, we designed a heavily engineered aerolysin nanopore, R282S/E216S/R220S/E222S/A260F, by substituting all four charged amino acids along the cis entry with short-chain uncharged serine to reduce the entry barrier of PFCA-R₆ complex, and introducing an additional cation- π interaction between AeL and R₆ probe to prolong the residence time of PFCAs in pore lumen. It was shown in SMD analysis that the entry barrier of C6 by 4S1F was 30% lower than R282A or 60% lower than WT (Fig. 48b), while the magnitude of exit barrier was maintained and its position was shifted towards the cis side, implying that rational engineering of nanopore interfaces held great potential to further improve the standard-free quantification and identification activity of single-molecule sensing of PFCAs. In particular, it was found that the energy barrier distributions of C6/C8/C10/C12/C14 were almost identical for R282S/E216S/R220S/E222S/A260F (Fig. S48c), which indicated that this heavily mutated nanopore would enable universal capture rates or calibration curves for all PFCAs, confirming the generality of standard-free quantification by single-molecule sensing.

Figure S48. (a) Required force to pull C0/C2/C4/C6/C8-R₆ conjugate through R282A aerolysin nanopore based on steered molecular dynamics (SMD). (b) Required force to pull C6-R₆ conjugate through WT, R282A, or R282S/D216S/R220S/D222S/A260F (4S1F) aerolysin nanopore based on SMD. (c) Required force to pull C6/C8/C10/C12/C14-R₆ conjugate through 4S1F aerolysin nanopore based on SMD.

2. I am concerned about the accuracy of the standard-free quantification in real environmental samples. Although the author has tested C2 in Figure 3c and S37, but how about medium and long chain PFCAs? In our previous comments, we also mentioned that the isomers of C2 are very limited and face little interference from environmental analogues. As the carbon chain grows, this problem theoretically becomes more and more serious. In Figure S37, the author tested all of untethered interferences for C2, how about tethered interferences, C3, C4, or haloacetic acid (difluoroacetic acid)?

Thank you for your questions. To demonstrate the excellent anti-interference ability of single-molecule sensing on other PFCAs, we carried out additional quantification measurements of C3 and C4 under either environmental or biological interference. Similar to our previous finding on C2, the calibration curves of C3 and C4 in the presence of tap water or serum (green and pink squares) were in good accordance with those measured in pure KCl electrolytes (black solid line), Fig. S43. These results emphasized once again that the untethered interferences had no influence on nanopore single-molecule sensing, and would not affect the performance of PFCA quantification, no matter whether the analytes were C2/C3/C4 or other median- and long-chain PFCAs.

Figure S43. Interference-free quantification of (a) C3 and (b) C4 in presence of tap water or serum at 20°C and -50 mV .

As for tethered non-PFCA interferences (i.e., mainly fatty acids or other species with a terminal carboxyl group), although they could generate non-negligible current signals, it was still possible to differentiate them from other PFCAs (Fig. S40); and even when the tethered interferences possessed indistinguishable current blockades or molecule volumes against PFCAs (e.g., FA4 and C3), it was simple to accurately identify them based on their dwell time or other signal features using clustering (Fig. S32). A customized density-based spatial clustering algorithm was developed and provided in this revision (more details were given in Workflow S2, and elaborated in the answer to Question 5). The number of signals identified for each species were very close to the ratios of their concentrations (Fig. S32), confirming that the tethered non-PFCAs interferences also would not affect either identification or quantification.

Figure S40. Quantification of C2-R₆ under tethered interference of valeric acid (FA5-R₆).

Figure S32. (a) One-, (b) two- and (c) three-dimensional distribution of cluster results of C3 and butyric acid (FA4), whose current blockades are extremely close. (d) Mean values of molecular volumes, $\Delta I/I_0$, I_σ , τ_{on} and H_{peak} of C3 and FA4. The concentration ratio of C3 and FA4 in mixed samples was set as 4:1, and the signal number ratio achieved by unsupervised clustering was 3.89:1.

With regard to the simultaneous quantification of multiple tethered PFCAs, we have performed a series of concentration-dependent measurements for various PFCAs-R₆ in the presence of internal standards (another PFCAs) with constant concentrations. Specifically, these measurements included the quantifications of 3H, 3Cl, FTA, or 7H with C5, 5H with C6, and C5 with C3 or C7 (Fig. S38), in addition to the previous quantification of C2 with C4, C5, or C6 (Fig. 3e, S35-S37). According to the theory of cumulative single-molecule captures rates for multiple PFCAs in dilute solutions, the ‘real’ interval times of internal standards should have remained constant since their concentrations didn’t change, as highlighted by the brown dashed line in each plot, while the R² values of the linear fitting for concentration-driven patterns of interval time were generally greater than 0.99 for target PFCAs. Moreover, the respective calibration curves matched well with those measured in pure or interfered electrolytes (Fig. S39). The above quantification measurements affirmed that, for WT AeL nanopore, the “one-calibration curve-fit-all” strategy covered at most to 5H, and when the carbon chain length increased from C5 to C7, the corresponding capture rates dropped quickly.

Figure S38. Simultaneous quantification instances of (a) 3H with C5-R₆ interference, (b) 3Cl with C5-R₆ interference, (c) 5H with C6-R₆ interference, (d) C5 with C3-R₆ interference, (e) C5 with C7-

R₆ interference, (f) FTA with C5-R₆ interference, and (g) 7H with C5-R₆ interference.

Figure 3. e Calibration curves of C2 under the interference of C4, C5 or C6. The concentration of C2 varied from 50 nM to 100 μM , while those of C4, C5 and C6 were kept at 6.7, 12.5 and 12.5 μM , respectively.

Figure S35. (a-d) Representative current traces (left) and signal intensity histograms (right) of C2-R₆ with different concentration (1~10 μM) under the interference of C4-R₆.

Figure S36. (a-e) Representative current traces (left) and signal intensity histograms (right) of C2-R₆ with different concentration (1~20 μM) under the interference of C5-R₆.

Figure S37. (a-e) Representative current traces (left) and signal intensity histograms (right) of C2-R₆ with different concentration (1~20 μM) under the interference of C6-R₆.

Figure S39. Calibration curves of C3, C4, 5H, C5 and C6 measured in the presence of internal standards of fixed concentrations.

3. I do not agree with the response of using different experimental conditions for qualitative and quantitative analysis, because standard free quantification requires standard free qualitative analysis first. Although the relationship between current and molecular volume or machine learning models can be reestablished, the slope of the linear relationship under different experimental conditions affects the resolution of qualitative analysis. Unless the author provides evidence to prove that the resolution of qualitative analysis is similar under the two conditions, I do not agree that quantitative analysis can be superior to LC-MS/MS and GC-MS. In addition, the two articles are also papers from 14 years ago.

Thank you for raising this critical question which allowed us to elaborate in more detail on the unique measuring principle and design strategy of nanopore single-molecule sensing. It is absolutely correct that standard-free identification is the prerequisite of standard-free quantification. However, according to the signal transduction mechanism of nanopore sensing, in particular those with double barriers, the identification and quantification performance were decided independently by either the exit or entry barrier. More specifically, the standard-free identification, including both standard-free prediction of current signals and high-resolution multi-feature classification, was determined by the electrostatic interactions between R₆ probe and positive-charge amino acids at the trans exit of AeL nanopore (e.g., K238 and K242). On the other hand, the standard-free quantification, i.e., to utilize a universal calibration curve for all PFCAs, was in our system hampered by the cis entry barrier into WT AeL nanopore (e.g., those four charged, long-chain amino acids, R282, D216, R220, and D222). In this revision, we have provided solid evidence to explain why it is possible to extend the standard-free quantification to medium- or long-chain PFCAs, as well as to continuously improve their LOD, without comprising the high-resolution standard-free identification capability.

As you mentioned, linear volume-current relationship was the core of our established standard-free identification and quantification method, and thus we reexamined carefully the current response of PFCAs measured with WT AeL at 30°C/-80 mV (Fig. S45a), R282A at 20°C/-40 mV (Fig. S45b), and R282A AeL at 30°C/-70 mV (Fig. 4g). Strikingly, it was found that the magnitude of blockades for PFCAs of various chain length (C2 to C9) always followed a linear pattern against their volumes in all three conditions, and shared an identical slope of 0.023% · Å⁻³ that was exactly the same as the value obtained by WT AeL at 20°C/-50 mV (Fig. 1d). Besides, it was also seen that the identification accuracy for C2 to C9 remained close to 100% by R282A at 30°C/-70 mV (Fig. S46). The constant slopes for linear volume-current relationship by R282A and WT AeL, no matter at a mild or elevated voltage and temperature, clearly proved that the standard-free prediction of signal features and high-resolution identification capability could be preserved, during the optimization process of standard-free quantification. This was because the positive-charge R₆ probe was able to drive various PFCAs to the same position within AeL pore lumen, due to the combined effect of electrophoretic force on R₆ probe and the electrostatic interactions between R₆ and K238/K242 residues at the trans exit of AeL nanopore. Hence, as long as the exit barrier of AeL was kept unchanged, both the linear volume-current relation and high-resolution identification would exist, enabling the decoupled enhancement of quantitative and qualitative analysis.

From a fundamental perspective, the slope of the linear volume-blockade relation was decided by the volume of sensing region based on the following equation, $\Delta I/I_0 = V_{analyte}/V_{sensing\ region}$. It was worth emphasizing that, for the big nanopore/small analyte combination (such as the aerolysin and PFCA-R₆ complex system in this study), the effective

transduction volume was smaller than the total size of pore lumen (e.g., 4.82 nm³ for WT AeL, which corresponded to only the region between A224 and S236 residues within β -barrel), and was highly sensitive to the actual residence location of analyte in nanopore. In this regard, the constant slopes in all conditions provided strong evidence that the probe-directed capture and sensing strategy was essential to ensure universal sensing region, and was crucial in this study to achieve standard-free quantification and identification of all PFCAs, via a series of nanopore interface modifications and temperature/voltage optimizations.

Figure S45. The established linear relationship between the hydrodynamic volume of C0-C9 PFCA-R₆ and the magnitude of their current blockade (a) by WT AeL at 30°C and -80 mV, and (b) by R282A AeL at 20°C and -40 mV.

Figure 4. g The linear volume-blockade relation measured with R282A at 30°C and -70 mV,

covering long-chain PFCAs up to C14.

Figure 1. d The established linear relationship between the hydrodynamic volume of C0/C2-C9 PFCAs-R₆ and the magnitude of their current blockade (blue squares and straight line). The volumes of PFCAs were calculated via MD simulation using GROMACS (Supplementary Fig. 2 and 3). The experimentally measured current blockade of 3H, 5H, 7H, 3Cl, and FTA was shown as brown circles, which was almost identical with the prediction from volume-current correlation (insert in Fig. 1d). The error bars on blue squares and brown circles were the sum of three times standard deviations (NOT one) obtained from the histogram of their current blockade.

Figure S46. The confusion matrix for identification of 8 PFCAs and R₆ probe by R282A AeL, at 30°C and -70 mV. Identification accuracy was 99.81%.

4. Although under the testing conditions of laboratory standards, the detection limit of C2 is lower than existing methods (HPLC-MS/MS and GC-MS). However, in the current HPLC-MS/MS method, the detection limit of C2 is the highest compared to other PFASs (0.1ng/L). Based on the existing results, we cannot speculate that the detection limit of other PFASs using single-molecule sensing will be better than the current HPLC-MS/MS method.

Thank you for raising this critical question. It is worth emphasizing that the proposed nanopore single-molecule sensors can achieve precise quantification and wide coverage simultaneously, while the current quantitative analysis techniques (such as HPLC-MS/MS) suffer from the severe shortage of authentic standards (less than 1% of the total PFAS). This underpins our vision to pursue standard-free quantification of PFCAs.

Nevertheless, LOD is clearly one of the key performance indicators when comparing any new detection method with existing instrumental analysis approaches. Therefore, taking into account the monitoring requirements of PFAS, our ultimate goal for the LOD of nanopore SMS is 1 pM, which corresponds to a LOD less than 1 ng/L for most PFCAs, e.g., 0.414 ng/L for C8, which is comparable to the performance of a recently reported FET sensor (*Nat. Water*, **2025**, DOI: 10.1038/s44221-025-00505-9). As shown in this revision, the R282A mutant AeL nanopore was able to reach a universal ‘LOD’ of 0.1 nM for C2 to C6 (Fig. 4d-4f, and S44b), and the R282S/E216S/R220S/E222S/A260F mutant pore could further reduce the entry barrier of C6, e.g., by 30% against R282A, extending the standard-free quantification to C14 (Fig. S48b and S48c). We had measured experimentally that this heavily engineered AeL nanopore was able to enhance the capture rates of median-chain PFCAs by 3-4 times (data not shown). In other words, the ‘true’ interval time of 30 pM PFCAs by 4S1F mutant at elevated voltage and temperature was ca. 15-20 seconds. Since the single-molecule identification performance of 4S1F was also significantly enhanced (e.g., ultralong second-level dwell time led to 0.01%-level standard deviation for blockade), a total number of 10 signals in 30 minutes was enough for quantification (*Nat. Nanotechnol.*, **2025**, 20, 523-531), which meant a ‘true’ interval time of 180 seconds, or potentially at least one more order of magnitude improvement in LOD. Note that another common and effective way to enhance the LOD of nanopore has yet to be adopted, i.e., utilizing an asymmetric electrolyte salt gradient to enlarge the capture regions of analytes, which is supposed to reduce the slope of interval time against analyte concentration as well (*Nat. Nanotechnol.*, **2025**, 20, 1079-1086), it is highly likely to realize a universal LOD of 1 pM for all PFCAs by 4S1F under the optimal quantification conditions (without compromising identification performance, as elaborated in detail above). In short, we have demonstrated experimentally and computationally that the R282A and 4S1F mutant can enhance the LOD of PFCAs by at least 5 times or 20 times, compared to WT, and sequentially expand the coverage of standard-free quantification to all PFCAs, which enables a competitive quantitative analysis performance (0.1-1 ng/L) with the state-of-the-art HPLC-MS/MS in the presence of authentic standards.

Figure 4. **d** SMD simulations for C0/C2/C4/C6-R₆ from cis entry to trans exit through R282A aerolysin nanopores. **e** Voltage-dependent capture rates of C0/C2/C3/C4/C5/C6-R₆ by R282A under the same concentration. **f** Calibration curve of C2, measured with R282A at 30°C and -70 mV. The concentration of C2 varied from 0.1 to 5 nM.

Figure S44. (b) Quantification of C4 by R282A AeL at 30°C and -70 mV to further optimize the limit of detection. The LOD was 0.1 nM or 21 ng·L⁻¹.

Figure S48. (b) Required force to pull C6-R₆ conjugate through WT, R282A, or R282S/D216S/R220S/D222S/A260F (4S1F) aerolysin nanopore based on SMD. (c) Required force to pull C6/C8/C10/C12/C14-R₆ conjugate through 4S1F aerolysin nanopore based on SMD.

5. The author provides a roadmap for constructing a machine learning model for standard-free identification. But the key to this approach lies in the ability of unsupervised clustering analysis to distinguish different PFCAs (workflow S2), which directly affects how much blockage or what type of PFCA for one single molecule signal is labeled as, and thus affects the training of machine learning models used for standard free qualitative analysis. Moreover, the author did not introduce this unsupervised clustering analysis, nor did they discuss its discriminative ability. I assume that if two groups of single-molecule signals with the same blockade are discovered through this unsupervised clustering analysis, how should we distinguish which PFCA they correspond to? This is very likely. According to Figure S41a provided by the author, we can observe that PFCAs with different molecular weights may have similar molecular volumes, although this figure mainly reflects the molecular volume differences of isomers of PFCAs with the same molecular weight. The reason why the machine learning model for standard-free identification in this article is so good, I speculate, is because the selected PFCAs have significant differences in blockade and can be completely separated by the unsupervised clustering analysis. Based on this, I am concerned that PFCAs or even non PFCAs that did not participate in the training of model may affect the accuracy of standard-free identification. And this concern can be eliminated by using PFCA standards.

Thank you for your comments. As explained above, the standard-free identification of PFCAs was composed of two components: 1) *standard-free prediction* of signal feature for unknown PFCAs based on the linear volume-blockade relationship; and 2) *high-resolution identification* of all PFCAs through multi-feature classification with in-house feature library. It was shown in Workflows 1 and 2 that the elementary steps in the ‘*Construction of feature library with known PFCA samples of high purity*’ and the ‘*Expansion of feature library with other PFCA samples in real samples*’ were similar, and the difference between those two processes were highlighted below in *italics* and/or **bold**:

Step a, Performing nanopore single-molecule measurements and analysis for *individual known PFCAs* (or *real samples with unknown PFCAs*) in the presence of internal standards to acquire raw current signals;

Step b, Extracting the frequency-modulated multi-dimensional features for *target PFCAs* (or *potential signals for unknown PFCAs*) and internal standards using customized script;

Step c, Applying the *two-dimensional kernel density plot analysis* (or *density-based spatial clustering algorithms*) to differentiate the signals of *target PFCAs* (or *unknown PFCAs*) and internal standards, (***and to primarily confirm the identity of unknown PFCAs based on the signal ratios of individual clusters and their appropriate abundance in real samples***);

Step d, (***Calculating the current blockade of cluster events induced by the unknown PFCAs, and utilizing linear volume-blockade relationship to estimate their respective molecular volumes***). Calibrating the multi-dimensional features of *target PFCAs* (or *unknown PFCAs*) against internal standards to reduce experimental errors. (***The identity of unknown PFCAs is re-affirmed with the MD simulated molecular volume library***);

Step e, (***Collecting and labeling the current signal induced by the identified unknown PFCAs, extracting the respective multi-dimensional features and including them into the feature library***). Using those labeled multi-dimensional features of *target PFCAs* (or *identified unknown PFCAs*) to train and select the appropriate classification models.

Workflow S1. Process of data acquisition, feature extraction and model training with known PFCAs.

Workflow S2. Process of standard-free identification and feature extraction for unknown PFCAs in real samples.

In this regard, the ultrahigh resolution classification and clustering abilities of nanopore sensors both originated from the multi-feature discriminability (in addition to blockade) of single-molecule current signals of PFCAs. To demonstrate this, we first examined in detail the clustering process of a mixed sample containing FA4 (butyric acid) and C3 (in a molar ratio of 4:1), which shared similar molecular volumes and current blockades. According to Workflow S2, we conducted nanopore SMS of this mixed sample and extracted multi-dimensional features from the single-molecule signals. We then applied OPTICS, a density-based unsupervised clustering algorithm, to separate the signals for FA4 and C3. In terms of clustering parameters, initial maximum distance between two samples was set as 7, and dynamic decay rate was set as 0.98. Three clusters were finally identified after multiple iterations, of which two were confirmed and labeled as FA4 and C3 according to the volume-current relationship and actual concentration ratio. We further investigated the spatial distribution of multi-dimensional features for these signals. Both clusters of signals shared an identical current blockade ($\Delta I/I_0$), indicating the signals of FA4 and C3 couldn't be separated or identified by one-dimensional feature (Fig. S32a). However, it was shown in the two-dimensional scatter plot of these signals that the spatial distance between those two clusters of FA4 and C3 was enlarged by adding a new feature I_{σ} (Fig. S32b). Moreover, in the three-dimensional space of I_{σ} , τ_{on} and H_{peak} , clustering effect of the signals induced by FA4 and C3 was significantly improved with a further enlarged spatial distance

(Fig. S32c). Most importantly, the ratio of clustered signals for FA4 to C3 (3.89:1) was close to their actual concentration ratio, confirming the feasibility to standard-free identify those single-molecule signals from structurally similar analytes with indistinguishable blockades. The molecular volumes, and mean values of $\Delta I/I_0$, I_σ , τ_{on} and H_{peak} for C3 and FA4 were summarized in Fig. S32d.

Figure S32. (a) One-, (b) two- and (c) three-dimensional distribution of cluster results of C3 and butyric acid (FA4), whose current blockades are extremely close. (d) Mean values of molecular volumes, $\Delta I/I_0$, I_σ , τ_{on} and H_{peak} of C3 and FA4. The concentration ratio of C3 and FA4 in mixed samples was set as 4:1, and the signal number ratio achieved by unsupervised clustering was 3.89:1.

Apart from the above cluster analysis of C3 and FA4, we would like to present more evidence for the superb identification capability of multi-feature classification, as shown in our recent studies on single-molecule sensing of PFCA aliphatic and aromatic isomers⁴⁹⁻⁵². For instance, seven typical aliphatic PFCA isomers were analyzed (Fig. S54), including two with four H/F substitution (Syn 28 and 1:3 FTCA), two with four H/F substitution plus one double-bond (4,5,5-3F and 4,4,4-3F), and three with two H/F substitution plus one oxygen addition (Syn 62, 2-OH, and R-2-OH). It was seen that the classification accuracy by blockade only was only 61.8% for all seven isomers (Figure S55), showing considerable mis-identification within and between different groups of isomers. In contrast, the overall classification accuracy of seven aliphatic could be increased to 99.4% by the frequency modulated 36-dimensional features (Figure S56). Another example was 17 H-substituted positional aromatic PFCA isomers (Figure S57c), including three with one H/F substitution (A2345/A2346/A2356), nine with two H/F substitution and one F/CF₃ substitution (B263/B243/B245/B264/B234/B345/B254/B354/B23456), and five with three H/F substitution and two F/CF₃ substitution (C24/C25/C26/C34/C35). The maximal peak separation for blockades was just 3.2% among 17 isomers (Figure S57d), leading to less than 50% classification accuracy for all three groups of isomers, and an overall accuracy of 31.1% for the total 17 isomers using blockade only (Figure S58a). This value

could be substantially improved to 90.4% using the frequency modulated 88-dimensional features (Figure S58f). Based on the above observations, it was clear that there were many other features of the single-molecule signals, other than blockade, being crucial to the standard-free identification of PFCAs, and the classification and clustering of various tethered PFCAs or non-PFCA interferences with similar blockade was not an issue at all.

Figure S54. Structures of typical PFCA aliphatic isomers: (1) 4,5,5-3F and 4,4,4-3F, (2) Syn 28 and 13FTCA, (3) Syn62, 2-OH and R-2-OH. [Adapted from: *Environ. Chem.* **2026**, 45, 1-15.]

		Accuracy/%						
Predicted label	2-OH					7.85	27.35	73.50
	R-2-OH			0.35	0.05	11.05	14.35	10.35
	Syn62			0.50	0.15	77.85	55.10	15.65
	13FTCA		2.70	48.30	81.00	0.05	0.15	
	Syn28		0.30	48.05	15.60	3.20	3.00	0.50
	4,5,5-3F	28.40	64.10	2.80	3.20			0.05
	4,4,4-3F	71.60	32.90					
		4,4,4-3F	4,5,5-3F	Syn28	13FTCA	Syn62	R-2-OH	2-OH
		True label						

Figure S55. Confusion matrix for the classification of 7 PFCA aliphatic isomers by blockade only. [Adapted from: *Environ. Chem.* **2026**, 45, 1-15.]

Figure S56. Confusion matrix for the classification of 7 PFCA aliphatic isomers by 36-dimensional features. [Adapted from: *Environ. Chem.* **2026**, 45, 1-15.]

Figure S57. (a) Schematic diagram of single-molecule electrochemical sensing. (b) Definition of features Blockade, Std and Dwell time. (c) Structural formulas and corresponding raw electrical signals of 17 H-PFCA aromatic isomers. (d-f) Blockade, Std and $\lg(\text{Dwell time})$ of 17 H-PFCA aromatic isomers calibrated by the internal standard C3. [Adapted from: *Environ. Chem.* **2025**.]

Figure S58. (a) Classification accuracy under SVM model within 1H, 2H, 3H, and 1H2H3H PFCA aromatic isomers, where abscissa 1 is *Blockade*, 2 is *Blockade + Std*, 3 is *Blockade + Std + lg(Dwell time)*, 19 is 19 quantiles, and 22 is the training result of the model using all features extracted from original signal. (f) Classification accuracy and confusion matrix of 1H, 2H, and 3H at 88 features. [Adapted from: *Environ. Chem.* **2025**.]

In summary, based on the current data of the author, I believe that the author can achieve the qualitative and quantitative analysis of C2 without standards in real environmental samples, and this process also requires the use of internal standards to reduce measurement errors, which is very similar to LC-MS/MS and GC-MS. However, if this is the case, the machine learning model used for standard free identification is not necessary, and only the linear relationship between molecular volume and blockade is needed to achieve the qualitative analysis of C2 without standard samples. These results still do not achieve significant progress.

Thank you for your insightful and valuable comments. Hope that, during this round of revision, we have provided sufficient evidence to support that the proposed nanopore single-molecule sensor is able to quantify multiple PFCAs simultaneously under real environmental conditions without the use of authentic standards. More specifically,

- a) we proved experimentally that R282A AeL was able to extend the linear volume-blockade relationship to C14 at -70 mV and 30°C (Fig. 4g), which meant that the signal of over 97% PFCAs could be correctly predicted without standards;
- b) we proved both experimentally and computationally that R282A AeL was able to standard-freely quantify PFCAs up to C6 (Fig. 4d-4e), covering almost 50% of the known PFCAs;
- c) we proved experimentally that the slope of volume-current relationship and identification capability of R282A AeL remained unchanged against WT, no matter at a mild or elevated voltage and temperature (Fig. 4g, S45-S46), confirming the feasibility to tune the strength of entry and exit barriers of AeL nanopore independently, and to continuously improve the standard-free quantification performance without compromising identification resolution;
- d) we demonstrated computationally that an extensively engineered 4S1F AeL would be able to standard-freely quantify PFCAs up to C14 (Fig. S48), covering over 97% of PFCAs;
- e) we proved experimentally that R282A AeL was able to extend the linear range of standard-free quantification of PFCAs to 0.1 nM (Fig. 4f and S44), or 11.4 ng • L⁻¹ for C2, among the best of all existing techniques;
- f) we analyzed theoretically that 4S1F AeL would be able to reach a universal LOD of 1 pM for all PFCAs under optimal quantification conditions, e.g., 10 signals in 30 minutes at the combination of high temperature, high voltage, and asymmetric electrolyte, comparable to the state-of-the-art HPLC-MS/MS in the presence of authentic standards;
- g) we proved experimentally that the calibration curves of (10) PFCAs were always identical no matter whether in the presence or absence of (any kind of) internal standards (Fig. 3e, S35-S39), confirming the feasibility to implement multiplexed digital quantification simply by adding up the individual capture rates;
- h) we proved experimentally that the excellent anti-interference capability of single-molecule sensing was applicable to other PFCAs as well (Fig. S43), confirming its reliability under a broad range of common environment or biological interferences, including tethered fatty acids (Fig. 3f, S32, and S40-S42);
- i) we proved experimentally that the multi-feature classification/clustering was able to standard-freely identify structurally similar PFCAs or interferences, even for those with indistinguishable molecular volumes or blockades (Fig. S32);
- j) we proved experimentally that the ultra-high resolution of PFCa classification/clustering originated from the multi-feature discriminability of single-molecule current signals (Fig.

2 and S50-S51), confirming the necessity of machine learning in the construction of feature library, and the enhancement of identification accuracy for standard-free quantification of all PFCAs.

Referring to the comments from Reviewer #3, 'although there remain some aspects to be solved for practical applications, ..., this work has fundamental value and brings a new perspective into a complex issue in analytical chemistry.' We hope the extra evidence provided in this round of revision can now fully address your concerns on the nanopore single-molecule sensor towards standard-free quantification of PFCAs. We are more than happy to present more information if you have any other questions. Look forward to receiving your comments and suggestions. Thank you.

Response Letter

Reviewer #1 (Remarks to the Author):

Review Comments:

While the authors have demonstrated good quantitative performance of their method with real samples, I wish to highlight that the current methodological framework cannot achieve truly "standard-free identification" of PFAS isomers.

Author Response:

Thank you very much for the recognition of our efforts in the last round of revision. It is greatly appreciated for your insightful comments to point out that "standard-free identification" of PFCA, especially for those structurally diverse isomers, has not been fully realized in the current work. As you suggested, *we have added one more limitation section at the end of discussion in this revision, to explicitly explain the difficulty of structural assignment for PFCA sharing similar volumes.* More importantly, we would also like to share with you our latest progress regarding (i) *the enhancement of blockade resolution for PFCA isomers*, and (ii) *the approach for standard-free prediction of other single-molecule signal features*, e.g., peak (H_{peak}), kurtosis (H_{kurt}), and standard deviation (I_{σ}), which demonstrates that the "truly" standard-free identification of PFCA isomers is possible for nanopore single-molecule sensors. More details are given below.

The results clearly show that blockade signal alone is insufficient for effective isomer differentiation:

1. "The classification accuracy by blockade only was only 61.8% for all seven isomers."
2. "The maximal peak separation for blockades was just 3.2% among 17 isomers (Figure S57d), leading to less than 50% classification accuracy... and an overall accuracy of 31.1%... using blockade only (Figure S58a)."

Consequently, when using Step d of Workflow 2 for PFCA structural identification, multiple possible PFAS matches will fall within the molecular volume prediction error range.

Author Response:

We fully agree with you that blockade only is insufficient for isomer identification, in particular when using WT aerolysin nanopore. To tackle this challenge, we have performed *extensive protein engineering to find out suitable aerolysin mutants that could substantially prolong the dwell time of PFCA inside nanopore* (unpublished), *reducing the full width at half maximum (H_{FWHM}) of current blockade*, and thus the volume prediction error as well. It was shown that *the classification accuracy for two same groups of 10 isomers could be considerably enhanced from 36.9% to 95.7%, using the blockade of WT and mutant aerolysin*, respectively. We admit that the sole improvement of blockade resolution could not "truly" achieve the standard-free prediction and identification of PFCA either, as the existence of PFCA or PFAS with indistinguishable molecular volumes is always inevitable, but it still represents a crucial step to strengthen the reliability of single-molecule signal clustering and identification for simultaneous determination of multiple targets in real environmental samples.

Figure S1. Confusion matrices for the identification of 10 PFCA isomers by (a) WT and (b) mutant aerolysin nanopore.

I acknowledge that supervised or unsupervised clustering algorithms can leverage "many other features of the single molecule signals, other than blockade" to distinguish different PFAS. However, the relationship between these features and molecular structures remains unclear. This means:

- These features can only be measured using known PFAS standards.
- Unlike molecular volume (which can be predicted from structure), these features cannot be computationally predicted for novel/unidentified PFAS structures.

Therefore, PFAS identification still fundamentally relies on:

1. The availability of PFAS standards for measurement, OR
2. Pre-existing databases built using such standards.

Both approaches are limited by the number of PFAS standards available.

Author Response:

Thank you very much for raising this highly valuable question. The standard-free prediction of other signal features (or more broadly speaking, *the standard-free construction of feature library for single-molecule signal identification*), although not fully achieved in the current work, *is indeed one of the key components towards standard-free quantification of all PFCAs*, and is the core objective of our ongoing work. We admit that the current version of workflow 2 still relies on certain minimal knowledge of the analytes, e.g., the approximate abundance of unknown PFCAs in real samples (do NOT have to be high-purity PFCA reference materials), so as to "standard-freely" expand the feature library. Therefore, for unknown PFCAs of indistinguishable molecular volumes, i.e., the resolution of predicted blockades might be insufficient for structural assignment, as discussed previously. The ultimate solution is to realize computational prediction for other features of single-molecule signals, just like the "volume-blockade" relationship established in this work. Below is our justification, and a brief description of our ongoing work, to demonstrate the feasibility of such prediction.

From a fundamental perspective, *the current fluctuations or features of single-molecule signals reflect the dynamic transitions among different "states" of individual analytes trapped in nanopore, and are decided by the interaction networks in between*. As for our "PFCA-R₆ / AeL" system, *since the residence position of R₆ probe inside AeL nanopore remains the same for various kinds of PFCAs*,

the respective interactions for PFCAs and the in-pore amino acid residues of AeL are unambiguous. In other words, the computational prediction of single-molecule signals or features purely based on the molecular structures of PFCAs is theoretically possible.

Note that we have already *experimentally established the single-molecule signal feature library for a total of 53 aromatic and 29 aliphatic PFCAs (including 16 classes of isomers)*, all resolved by multi-feature clustering, this allows us to *investigate the change of features by elementary reactions in PFCA transformation using deep learning*. For instance, we have quantified the influence of four elementary reactions during the conversion of fluorinated aromatic carboxylic acids (analogues and positional isomers), i.e., H/F substitution, F/CF₃ substitution, CF₂ addition, and CH₂ addition. Based on these finding, the most important signal features for identification, e.g., blockade ($\Delta I/I_0$), standard deviation (I_σ), peak (H_{peak}), and kurtosis (H_{kurt}), have been successfully predicted (in the absence of standards) for six fluoro-(trifluoromethyl)phenylacetic acid isomers (unpublished, data not shown). Physical origins of the above single-molecule signal features can also be well characterized through all-atom molecular dynamic simulations, enabling the combination of computational simulation and generative modelling in the future for ‘standard-free’ construction of feature library for all PFAS.

In short, the broader sense of “structure-feature relationship” suggests that the identification of PFCAs by single-molecule sensing could completely avoid the use of reference standards. However, it has not been fully realized in the current manuscript. Hence, we have *added two more paragraphs at the end of discussion section, to explain the limitations of current work and the future perspectives on mitigation approaches*.

Figure S2. Schematic illustration of standard-free prediction of fluoro(trifluoromethyl)phenylacetic acid (CH₂+1F/1CF₃) isomers’ signal features using four elementary reactions. The chemical formula of eight classes of fluorinated aromatic carboxylic acids is denoted as “X + mF / nCF₃”, where “X” describes the functional groups at position 1 and can be “CH₂”, “CF₂” or “0”, while “mF / nCF₃” is the functional groups at positions 2~6 ($m + n \leq 5$). For example, “0+1CF₃” contains three isomers of (trifluoromethyl)benzoic acids.

The current method primarily uses the predicted molecular volume database for final structural assignment. This creates a significant limitation: identification is confounded by PFAS sharing similar molecular volumes. The authors' own example distinguishing FA4 and C3 illustrates this:

- If FA4 were an unknown PFAS with a similar volume, the method could only determine it is not

C3. Its specific identity would depend entirely on whether a matching structure exists in the predicted volume database.

- While the FA4/C3 case might be hypothetical, the potential for confusion is demonstrably real for the medium/long chain PFAS isomers studied in the authors' previous work (e.g., the 17 H-substituted positional aromatic PFCA isomers or the seven typical aliphatic PFCA isomers). In those studies, the authors had standards and successfully used clustering to differentiate them. Crucially, if some of these were unknowns, Workflow 2 could not definitively assign their structures due to volume similarities.

Author Response:

We greatly appreciate and fully agree with your critical observations. It is true that the volume-determined structural assignment could not fully solve the challenge for standard-free identification of PFCA isomers sharing indistinguishable volumes. Therefore, following your suggestion, we have explicitly stated the limitation of current method in the revised discussion that the molecular volume similarity may result in ambiguity during structure assignment, particularly for PFAS families with dense structural space. Moreover, we have also offered two possible improvement approaches in the perspective section, in order to enhance the blockade resolution for PFCA isomers, and to standard-freely predict other features of single-molecule signals.

Recommendation: Given these limitations, the authors should refrain from emphasizing "standard-free identification" as a key capability of the current method. The dependence on standards (either directly or via pre-built databases) and the inherent limitation posed by similar molecular volumes should be explicitly stated in the manuscript's limitations section.

Author Response:

We highly appreciate your constructive recommendation and have thus included two additional paragraphs for limitations and future perspectives at the end of discussion section. Please see below for details.

“Regarding the limitations of the current method, it should be noted that the volume-dependent structural assignment principle may encounter difficulties in standard-free identification, especially when dealing with “unknown” (i.e., those yet to be included in the feature library) PFCA analogues or isomers of indistinguishable volumes. In this case, the current Supplementary Workflow 2 would still require a certain amount of prior knowledge, e.g., on the relative abundance of unknown PFCAs in real samples, so as to avoid the use of high-purity PFCA reference materials for the construction of single-molecule signal feature library. Without these, the insufficient blockade resolution of WT AeL nanopore could not unambiguously identify structurally diverse PFCA isomers in the absence of standards.

Looking forward, two research directions may mitigate the above limitations. One is to design suitable aerolysin mutants that could prolong the dwell times of PFCAs inside nanopore, narrow the distribution of blockades, reduce their prediction or identification error, and eventually enhance the resolution for PFCA isomers of similar volumes. The other is to computationally predict additional nanopore signal features, leveraging the pre-established and fully resolved feature

database, as well as the precisely controlled probe-nanopore interactions. Advances in deep learning, MD simulations, or generative modelling will support the formation of a more broadly structure-feature relationship. These potential improvements, while beyond the scope of the present work, illustrate possible paths for extending the utility of single-molecule sensing in PFAS analysis.”

Response Letter (4th Round Peer Review)

Reviewer #1 (Remarks to the Author):

Review Comments:

The authors have done a good job addressing the revisions. The limitations of the method have been effectively supplemented, and claims regarding the "free-standard identification" have also been appropriately restricted to the pre-established database. I have no further comments.

Author Response:

Thank you for your recognition of our work! It is greatly appreciated for your continuous support to improve the quality of our work.